# Bilateral trade potential analysis of the Lanzhou-Kathmandu South Asian rail-road freight trains linking China and Nepal: A stochastic frontier gravity model approach

**Fei Tian** [ID]*

China Railway First Survey and Design Institute Group Co., LTD, Xi'an Shanxi, China

\* 736827849@qq.com

## Abstract

In this paper, the stochastic frontier gravity model is applied to analyze the trade potential between China and Nepal and the prospects of Lanzhou-Kathmandu South Asian rail-road freight trains (LKSARFT). Based on the statistical data, we test the Exports Efficiency (EE), Bilateral Trade Efficiency (BTE), Exports Trade Potential (ETP), Bilateral Trade Potential (BTP), Extended Exports Trade Potential (EETP), Extended Bilateral Trade Potential (EBTP), Improved Exports Trade Potential (IETP) and Improved Bilateral Trade Potential (IBTP) between China and Nepal, the following analysis results can be found: for the bilateral trade model, the bilateral non-efficiency factor decreasing at a rate of 0.057 with time increasing, bilateral trade increasing at a rate of 0.057 with time increasing. For the exports model, the exports non-efficiency factor increasing at a rate of 0.004 with time increasing, exports trade decreasing at a rate of 0.057 with time increasing. The BTE between China and Nepal increases when time changes, the EE from China to Nepal remains constant changing during the 18 years. The changing range of BTE is 0.002–0.05; the changing range of EE from China to Nepal is over 0.1, larger than the BTE. The BTE and EE ranking among the eight South Asian countries are ranking fifth and fourth during the 18 years. exports trade resistance from China to Nepal is larger than bilateral trade resistance; The import trade potential from Nepal to China is huge, the focus of bilateral trade between China and Nepal may be changed, there are more goods may be exported from Nepal to China, and China may become trade deficit when trading with Nepal. Then, the development bottlenecks of the LKSARFT are analyzed. Finally, we give policy directions to boost bilateral trade efficiency and tap the potential of bilateral trade between China and Nepal.

## 1. Introduction

The "the Belt and Road" (B&R) proposal, including "the Silk Road Economic Belt" and "the 21st-Century Maritime Silk Road", was firstly proposed by China in 2013 [1, 2]. The B&R is China's greatest international economic ambition, which focuses on stimulating economic

**Funding:** This research was supported by the National Natural Science Foundation of China (71173177). The authors would like to thank the anonymous referees for their valuable comments and suggestions.

development in a vast region covering sub-regions in Asia, Europe and Africa, accounting for 64% of world population, over 40 countries and 30% of world GDP. The B&R is devised to reconfigure China's external sector in order to continue its strong growth. While infrastructure development plays a central role, the B&R belongs to a comprehensive initiative including policy dialogue, unimpeded trade, financial support and people-to-people exchange. In November 2013, this initiative was written into the comprehensive reform blueprint adopted by the Party leadership as a key policy priority before 2020 [3]. In March 2015, with approval by the State Council, several government departments jointly laid out detailed plans for the B&R [4].

Some research papers were also devoted to the B&R, [5] provided answers to some basic questions including the exactly nature of B&R, the China's motivation on B&R, and the possible impact on the existing economic order. [6] raised three questions, the first question was the real objectives behind the B&R, the second question focused on whether the B&R was driven by market-based transactions or was a form of foreign aid that was not based on economic calculation of gains and losses, the third one went to the economic cooperation priority targets of the 60 or some countries in Asia, Europe and Africa along the Belt and Road. In order to seek joint establishment of "the 21$^{st}$-Century Maritime Silk Road" is an important strategy to promote the new round of China's opening up and to realize common development with countries along the Road, and the trade is a foundation and vital link of this strategy. [7] proposed the use of Markov chains to forecast time-varying logistic distribution flows for a three-layer supply chain framework. [8] modified the gravity prediction model to calculate the changes in transshipment traffic. [9] explored a bi-level programming model to reconstruct the shipping service network between Asia and Europe. [10] discussed the research trends and agenda on the Belt and Road initiative with a focus on transportation and logistics. [11] proposed a method for the selection of the most urgent need for transnational high-speed railway construction in the B&R region. [12] applied a global computable general equilibrium model to investigate the macroeconomic impact of China's Belt and Road Initiative. [13] explored the supply chain coordination issues arising from the B&R, and investigated the impacts of the cost sharing contract on the key decisions for logistics service supply chain with mass customization. [14] thought the Central and West Asia, Western Europe and Russia are favorable destinations of Chinese overseas direct investment (ODI). [15] estimated trade potential and trade efficiency of countries along the Maritime Silk Road with each other using the Stochastic Frontier Gravity Model. They found that trade efficiency of the Maritime Silk Road appeared on an increasing trend, and China still had great potential in terms of exports growth with other countries along the Road. Meanwhile, they suggested that China should accelerate its regional economic integration, reduce tariff and non-tariff barriers, improve the level of trade facilitation, increase maritime connectivity, promote transport infrastructure, and strengthen cooperation on the prevention of financial risks. By applying the same Stochastic Frontier Gravity Model, [16] estimated the China's outward foreign direct investment (FDI) efficiency and determined in 69 countries along the Belt and Road over the period of 2003–2013. They found China's outward FDI was significantly restricted by some man-made barriers in host countries; China had huge outward FDI potential in countries along the Belt and Road. Now China is just promoting the B&R steadily, and some projects have been achieved, for example, the Lanzhou-Kathmandu South Asian rail-road freight trains (LKSARFT).

There are eight countries in South Asian area, with total 1.56 billion people and huge free trade market demand. The economic and cultural communication between China and other South Asian countries only via sea transportation because of the Himalayas since ancient times, but with the fast development of the railway and road infrastructure, China decided to develop the land routes to connect the South Asian countries. As a positive response to B&R,

the government of Lanzhou began to operate the Lanzhou-Kathmandu South Asian rail-road freight trains (LKSARFT) in May 11, 2016. Lanzhou becomes the first inland city in China to operate the South Asian freight trains. The LKSARFT provide a land trade connection between Lanzhou, Gansu Province of China and Kathmandu, the capital of the Federal Democratic Republic of Nepal. LKSARFT is a very minor corridor in terms of trade volume and commodity structures, compared to other corridors of B&R, but when we have our field research in Lanzhou, we found that a lot of business men sold their goods between China and Nepal via LKSARFT, it is very important for Chinese strategic considerations in South Asia.

LKSARFT belongs to railway and road international and intermodal transportation logistics service, using containers to carry the freight. The transported cargos are light industrial products, food, medicine, building materials, cement and so on. Each container's weight is about 10 to 20 tons; each train carries 35 containers at most because of the railway traction limitation in the Golmud-Lhasa railway section. For each container, the transport charging including two parts: freight owners cost 30000 CNY per container, the government provides another 10000 CNY per container. 7 days are needed to transport the cargos from Lanzhou to Kathmandu, while traditional sea and road international transportation need 35 days. The whole transport processes of LKSARFT included three parts (see Fig 1): (i) From Lanzhou International Port Area to Shigatse, China, passing through the Lanzhou-Qinghai Railway and Qinghai-Tibet Railway, the total operation length is 2431 kilometers and the running time is about 3 days; (ii) From Shigatse, China to Geelong town, China, through the roadway, the total operation length is 540 kilometers and the running time is about 2 days; (iii) From Geelong town, China to Kathmandu, Nepal, through the roadway, the total operation length is 184 kilometers and the running time is about 2 days. Compared with the traditional transportation mode, LKSARFT is more time-saving and safe. See [17].

The South Asian country, Federal Democratic Republic of Nepal, which is located in the south of the Tibet, China and north of India, with total 27.6 million people. Nepal is an agricultural-based developing country; about 80% people are engaged in agriculture, 16% population work for the service industries and only 4% in industry. By the end of 2015, Nepal's agriculture accounting for 32.71% of GDP, service industries accounting for 40.83% of GDP, industry accounting for only 15.6% of GDP, and the other is 10.9% [18]. Nepal's industry is still in the early development stages with weak foundation and infrastructure, small scale of production and low level of mechanization. The government of Nepal classifies industry into four types, including manufacturing industry, energy-based industry, mining industry and construction industry. The products of the manufacturing industry includes food, beverages, tobacco, textiles, leather products, shoes, wood products, paper and paper products, chemical products, rubber products, plastic products, building materials, iron products, tools and electrical products etc. The national statistical data [18] shows that: (i) Nepal imports 57 million tons food during 2015; (ii) Nepal has 9 cotton textile enterprises, 6 chemical fiber products enterprises and 12 hemp products enterprises, the annual textiles demand is about 300 million meters, but the actual production capacity is less than 7 million meters; (iii) Nepal's chemical products only including soap, washing powder and match; (iv) Western medicine demand in Nepal is about 10 billion rupees for each year, domestic produced western medicine can only meet 5%

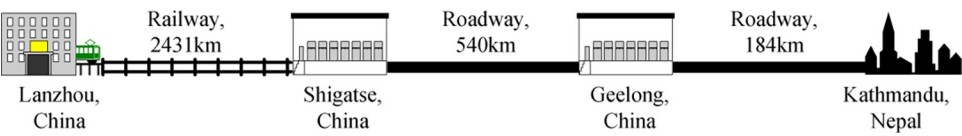

**Fig 1. The whole transport process of LSSARFT.**

to 10% demand, others depends on imports from other countries; (v) Cement demand in Nepal is about 10 billion for each year, domestic produced cement can only meet 40% demand, others depends on imports from other countries; (vi) Nepal's wood carvings, bronze carvings, stone carvings, carpets, tapestries, blankets, weavings, embroidery and leather clothing are high quality and cheap, most of these are exports freights. According to the statistical results we can find that Nepal has huge import demand in food, textiles, building materials and medicine etc. China has strong light industry production capacity; the bilateral trade complementarity is strong between Nepal and China. And now, the LKSARFT provides a better opportunity for bilateral trade potential between China and Nepal. In this paper we focus on studying the prospects about LKSARFT as well as bilateral trade potential between China and Nepal based on stochastic frontier gravity model (SFGM), quantitatively.

The remainder of this paper is organized as follows: In section 2, the prospects about LKSARFT and bilateral trade potential between China and Nepal based on the SFGM are studied. First, a detail introduction about the previous theoretical researches about SFGM is give, then the needed initial data used to formulate the exports model and bilateral trade model is introduced, next the final calculation results and related analysis are given. In order to ensure the development sustainability of LKSARFT, section 3 is devoted to analyze the development bottleneck and some of our advisement for the LKSARFT based on the field research of the LKSARFT. The final section 4 presents the major conclusions and gives an outline of future research task.

The advantages of studying the LKSARFT development prospects by applying SFGM are as follows: (i) SFGM does not suffer from loss of estimation efficiency. (ii) SFGM corrects for the economic distance bias term, which is creating non-normality, isolating it from the statistical error term. (iii) The suggested approach provides potential trade estimates that are closer to frictionless trade estimates, because the approach represents the upper limits of the data, which come from those economies that have liberalized their trade restrictions the most [19]. (iv) The SFGM bears strong theoretical and trade policy implications towards finding ways of minimizing unilateral impacts to volume of trade.

## 2. Literature review

### 2.1 Bilateral trade potential: Based on SFGM

Trade across regions and borders are considered important in improving welfare of people [20, 21]. The B&R initiative provides good bilateral development opportunities for both China and Nepal, as well as the other seven South Asian countries. The LKSARFT acts as one of the important land belts, but how about the trade potential of the two countries? What factors promote or limit bilateral trade between the two countries? How to improve bilateral trade efficiency and tap the potential of bilateral trade between the two countries? All these questions' quantitative study results are helpful to improve the development of the LKSARFT, as well as the development of the B&R. Next these questions are discussed based on stochastic frontier gravity model in detail.

### 2.2 The stochastic frontier gravity model

Some early literatures have estimated the difference between observed values and the estimated predicted values, by using an augmented gravity equation through Ordinary Least Squares (OLS) to assess the performance of bilateral potential trade among a pair of countries [22–25]. The OLS estimation procedure produces estimates that represent the centered values of the data set. However, potential trade refers to free trade with no restrictions to trade, some non-efficiency trade factors can't be observed through this model, so for policy purposes, it is

rational to define potential trade as a maximum possible trade that can occur between any two countries, which has liberalized trade restrictions the most, given the determinants of trade. This means that the estimation of the potential trade requires a procedure that represents the upper limits of the data and not the centered values of the data [26]. To solve this problem, [27] proposed the concept of stochastic production frontier analysis which deals with the upper bound of the data set to measure the maximum possible output, this approach is known as the Stochastic Frontier Gravity Model (SFGM). The SFGM is the Integration of Gravity Model and Stochastic Frontier Production Function Model which was formally introduced by [28] to address the inherent bias of the conventional gravity model of trade and to estimate potential trade flows. [29] applied frontier production function analysis to small farms in Nicaragua during 1998–2005, the results showed an acceptable average of technical efficiency which the makers of public policy in Nicaragua a must consider for the future. [30] focused on examining Philippines's exports efficiency and potential based on trading partner's characteristics using SFGM, unlike the usual measure of gravity model using OLS that measure potential from the mean. [31] provided a survey about environmental efficiency measure issue. the paper deals with different ways for including environmental variables, which offered several perspectives for measuring efficiency with frontier methods. [15] estimated the potential and efficiency of trade flows between China and countries along the Maritime Silk Road from 2005 to 2013 by using the SFGM. Furthermore, [16] defined the extent of the Belt and Road in terms of geographical boundaries, justified the application of the SFGM to the foreign direct investment (FDI) analysis, and constructed a frontier regression model to assess the China's outward FDI efficiency. [32] used an extended gravity model to examine the determinants, efficiency and potential of agri-food exports from Nigeria to the EU for the 1995–2019 period the study showed that Nigeria's agri-food exports with the EU has a relatively large potential that has not been exploited. The stochastic frontier gravity model is applied on a dataset including 35 countries during 2009–2017 by [33], the results indicated that the trade resistance of China's export to countries along the Belt and Road has increased over time, while there is still huge trade potential at various industries. [34] provided empirical insights on the determinants and potential of agri-food exports from Nigeria to 70 major trading countries between 1995 and 2019 by applying a Stochastic Frontier Analysis on a gravity model. [35] aimed to examine the key determinants and efficiency of China's agricultural exports with its 114 importing countries by applying the Stochastic Frontier Analysis on an augmented gravity model for the period of 2000–2019.

## 3. Methodology of the study

In this paper we will use the SFGM to exam the bilateral trade potential between China and Nepal. With a stochastic frontier approach, the basic gravity equation can be written as:

$$Y_{ijt} = f(X_{ijt} \cdot \alpha) \cdot e^{v_{ijt} - \mu_{ijt}} \tag{1}$$

$$Y_{ijt}' = f(X_{ijt} \cdot \alpha) \cdot e^{v_{ijt}} \tag{2}$$

$$TE_{ijt} = Y_{ijt}/Y_{ijt}' = e^{-\mu_{ijt}} \tag{3}$$

$$\mu_{ijt} = [e^{-\eta(t-T)}] \cdot \mu_{ij} \tag{4}$$

For sample $i$, $j$ (e.g. two different countries) and the calculation period $t$; $Y_{ijt}$ is the actual output from country $i$ to country $j$; $Y_{ijt}'$ is the optimal output from country $i$ to country $j$; $f$

$(X_{ijt} \cdot \alpha)$ is a function of the determinants of potential trade input variables $X_{ijt}$; the parameter to be estimated $\alpha$; $v_{ijt}$ represents the random disturbance, $v_{ijt} \sim \text{iidN}(0, \sigma^2)$, captures the influence on trade flows of other left out variables, including measurement error that are randomly distributed across observations in the sample; $\mu_{ijt}$ represents the non-efficiency trade parameter, which is due to the influence of the behind the border measures of the importing country, this bias creates the difference between actual and potential trade between two countries [36]; $\mu_{ijt}$ takes value between 0 and 1 and it is usually assumed to follow a truncated (at 0) normal distribution. When $\mu_{ijt}$ takes the value 0, this indicates that the bias or country-specific beyond the border constraints are not important and the actual exports and potential exports are the same, assuming there are no statistical errors. When $\mu_{ijt}$ take the value other than 0 (but less than or equal to 1), this indicates that the bias or country-specific beyond the border constraints are important and they constrain the actual exports from reaching potential exports; $TE_{ijt}$ is the technical efficiency. The Eq (4) shows that the non-efficiency trade parameter changes when the time changes; $T$ is the total calculation period number; $\mu_{ij}$ belongs to truncated normal distribution. If parameter to be estimated $\eta = 0$, shows the non-efficiency trade parameter wouldn't change when the time changes; if $\eta > 0$, shows the non-efficiency trade parameter decreasing when the time changes; and if $\eta < 0$, shows the non-efficiency trade parameter increasing when the time changes. Next we change the Eq (1) into logarithmic form:

$$\ln Y_{ijt} = \ln f(X_{ijt} \cdot \alpha) + v_{ijt} - \mu_{ijt}, \mu_{ijt} \geq 0 \tag{5}$$

Based on Eq (5), we replace the output with trade volume and reference the related input parameters used in the traditional SFGM [37–41]. We establish two regression equations, including exports regression equation model and bilateral trade regression equation model, to formulate the trade potential and trade efficiency. Shows as follows:

$$\ln\text{EXP}_{ijt} = \alpha_0 + \alpha_1 \ln\text{PGDP}_{it} + \alpha_2 \ln\text{PGDP}_{jt} + \alpha_3 \ln\text{P}_{it} + \alpha_4 \ln\text{P}_{jt} + \alpha_5 \ln\text{D}_{ij} + \alpha_6 \text{B}_{ij} + \alpha_7 \text{LAN}_{ij} \\ + \alpha_8 \text{LANG}_{ij} + \alpha_9 \text{FTA}_{ij} + v_{ijt} - \mu_{ijt} \tag{6}$$

$$\ln\text{EAI}_{ijt} = \alpha_0 + \alpha_1 \ln\text{PGDP}_{it} + \alpha_2 \ln\text{PGDP}_{jt} + \alpha_3 \ln\text{P}_{it} + \alpha_4 \ln\text{P}_{jt} + \alpha_5 \ln\text{D}_{ij} + \alpha_6 \text{B}_{ij} + \alpha_7 \text{LAN}_{ij} \\ + \alpha_8 \text{LANG}_{ij} + \alpha_9 \text{FTA}_{ij} + v_{ijt} - \mu_{ijt} \tag{7}$$

Where $i$ and $j$ represent the two counties; $\text{EXP}_{ijt}$ represents the exports volume from country $i$ to country $j$; $\text{EAI}_{ijt}$ is the bilateral trade volume between country $i$ and country $j$; $\text{PGDP}_{it}$ and $\text{PGDP}_{jt}$ represent the per capita gross domestic product in the two counties; $\text{P}_{it}$ and $\text{P}_{jt}$ represent the total population in the two countries; $\text{D}_{ij}$ is the distance between the two countries. According to the requirements of the research object, it can be the straight line distance (or transport line distance) between capitals in two countries, or shipping distance between major ports in two countries, or straight line distance (or transport line distance) main trading cities in two countries; $\text{B}_{ij} = 1$ shows there is a common border between the two countries, otherwise $\text{B}_{ij} = 0$; $\text{LAN}_{ij} = 1$ shows both the two countries are landlocked countries, otherwise $\text{LAN}_{ij} = 0$; $\text{LANG}_{ij} = 1$ shows the two countries share the same language, otherwise $\text{LANG}_{ij} = 0$; And $\text{FTA}_{ij} = 1$ shows there is a free trade agreement in force between the two countries, otherwise $\text{FTA}_{ij} = 0$. The variables in this paper are presented in Table 1.

In order to test whether the Eq (6) and Eq (7) are suitable to assess the performance of bilateral potential trade among a pair of countries, we can use Maximum Likelihood Estimate (LR) to test the model. We should test two aspects including whether the non-efficiency trade existed, and whether the non-efficiency trade changes when time changes. The tests will work

**Table 1. The variables.**

| Variable | Explanation | Variable | Explanation |
|---|---|---|---|
| $EAI_{ijt}$ | The bilateral trade volume between country $i$ and country $j$ | $EXP_{ijt}$ | Exports volume from country $i$ to country $j$ |
| $PGDP_{jt}$ | The per capita gross domestic product in country $i$ | $P_{it}$ | The total population in country $i$ |
| $PGDP_{it}$ | The per capita gross domestic product in country $j$ | $P_{jt}$ | The total population in country $j$ |
| $D_{ij}$ | The distance between the two countries | $B_{ij}$ | Situation of border between the two countries |
| $LAN_{ij}$ | The type of land between the two countries | $LANG_{ij}$ | Type of language between the two countries |

as follows: Assume $H_0:\gamma = \mu = \eta = 0$ and $H_0:\eta = 0$. Calculate LR Statistics value of Log likelihood under unconstrained and constrained conditions, respectively. Compare LR Statistics value with the critical value of chi-square distribution with 1% significance level. Refuse the assumptions if LR Statistics value is larger than critical value, otherwise accept the assumptions.

### 3.1 The needed initial data

In this paper we try to use SFGM to study the performance of bilateral potential trade between China and Nepal, this model needs to collect a lot of regional related countries' data. In this paper, we compared and analyzed eight South Asian countries' data from 2001 to 2018; there are Nepal, India, Pakistan, Bangladesh, Afghanistan, Bhutan, Sri Lanka and Maldives. The needed data was showed in Tables 2 and 3 [42].

In this paper we study the LKSARFT, so $D_{ij}$ are the straight line distances between Lanzhou to the capital city of other related countries: Lanzhou-Kathmandu, Lanzhou-New Delhi, Lanzhou-Islamabad, Lanzhou-Dhaka, Lanzhou-Kabul, Lanzhou-Thimphu, Lanzhou-Colombo, Lanzhou-Male.

The results presented in Table 2 demonstrate a steady increase in the bilateral trade volume between China and the eight South Asian countries, as well as China's export volume to these nations from 2001 to 2018. Notably, the bilateral trade volume and export amount between China and India was the highest among the eight countries. The rankings of bilateral trade volume and export amount between China and the South Asian countries from one to eight are as follows: India, Pakistan, Bangladesh, Sri Lanka, Nepal, Afghanistan, Maldives, and Bhutan. Furthermore, the ranking of China's export amount to the eight South Asian countries from one to eight is also as follows: India, Pakistan, Bangladesh, Sri Lanka, Nepal, Afghanistan, Maldives, and Bhutan. The subsequent discussion will primarily focus on Nepal.

## 4. Results and discussion

### 4.1 The results and analysis

First of all, we should test whether the Eq (6) and Eq (7) are suitable to assess the performance of bilateral potential trade among the countries mentioned above. The testing results were presented in Table 4. We refused the assumptions because all LR Statistics values were larger than critical value, which means the two model are suitable to describe the non-efficiency trade, and the non-efficiency trade changes when time changes.

We used Frontier 4.1 software to regression analyze the initial data mentioned in Tables 2 and 3. The basic regression analysis and the model checking results were presented in Table 5. The trade efficiency analysis results between China and Nepal were presented in Table 6, including exports efficiency (EE) and bilateral trade efficiency (BTE), as well as the rankings of EE and BTE among the eight South Asian countries. The trade potential measurement results between China and Nepal were showed in Table 7.

**Table 2. The needed initial data of the eight South Asian countries (from 2001 to 2018).**

| | Year | $EAI_{ijt}$ (Billion USD) | $EXP_{ijt}$ (Billion USD) | $PGDP_{it}$ (USD) | $PGDP_{jt}$ (USD) | $P_{it}$ (Billion) | $P_{jt}$ (Billion) |
|---|---|---|---|---|---|---|---|
| $i$: China $j$: Nepal | 2001 | 1.532 | 1.486 | 1041.64 | 240.47 | 12.72 | 0.250 |
| | 2002 | 1.103 | 1.051 | 1135.45 | 236.71 | 12.80 | 0.256 |
| | 2003 | 1.273 | 1.220 | 1273.64 | 242.14 | 12.88 | 0.261 |
| | 2004 | 1.715 | 1.632 | 1490.38 | 272.25 | 12.96 | 0.267 |
| | 2005 | 1.964 | 1.879 | 1731.13 | 298.01 | 13.04 | 0.273 |
| | 2006 | 2.680 | 2.598 | 2069.34 | 326.04 | 13.11 | 0.287 |
| | 2007 | 4.000 | 3.860 | 2651.26 | 362.22 | 13.18 | 0.284 |
| | 2008 | 3.810 | 3.750 | 3413.59 | 434.96 | 13.25 | 0.290 |
| | 2009 | 4.140 | 4.090 | 3748.93 | 438.29 | 13.31 | 0.294 |
| | 2010 | 7.437 | 7.322 | 4432.96 | 534.52 | 13.38 | 0.300 |
| | 2011 | 11.950 | 11.810 | 5444.79 | 619.45 | 13.44 | 0.305 |
| | 2012 | 19.980 | 19.680 | 6188.19 | 706.65 | 13.51 | 0.275 |
| | 2013 | 22.540 | 22.100 | 6807.43 | 694.10 | 13.57 | 0.278 |
| | 2014 | 23.300 | 22.830 | 7593.88 | 696.94 | 13.64 | 0.282 |
| | 2015 | 28.660 | 27.890 | 7924.65 | 732.30 | 13.71 | 0.285 |
| | 2016 | 30.245 | 28.900 | 8000.34 | 756.44 | 13.72 | 0.299 |
| | 2017 | 32.335 | 31.008 | 8123.22 | 788.00 | 13.98 | 0.310 |
| | 2018 | 35.667 | 33.450 | 8256.77 | 812.22 | 14.31 | 0.334 |
| $i$: China $j$: India | 2001 | 35.960 | 18.960 | 1041.64 | 459.58 | 12.72 | 10.71 |
| | 2002 | 49.460 | 26.710 | 1135.45 | 480.21 | 12.80 | 10.89 |
| | 2003 | 75.950 | 33.430 | 1273.64 | 558.44 | 12.88 | 11.06 |
| | 2004 | 136.040 | 59.360 | 1490.38 | 642.56 | 12.96 | 11.23 |
| | 2005 | 187.030 | 89.340 | 1731.13 | 731.74 | 13.04 | 11.40 |
| | 2006 | 248.600 | 145.810 | 2069.34 | 820.30 | 13.11 | 11.57 |
| | 2007 | 386.500 | 240.110 | 2651.26 | 1055.14 | 13.18 | 11.74 |
| | 2008 | 518.400 | 315.850 | 3413.59 | 1027.91 | 13.25 | 11.91 |
| | 2009 | 433.830 | 296.560 | 3748.93 | 1126.95 | 13.31 | 12.08 |
| | 2010 | 617.610 | 409.150 | 4432.96 | 1375.39 | 13.38 | 12.25 |
| | 2011 | 739.080 | 505.370 | 5444.79 | 1488.52 | 13.44 | 12.41 |
| | 2012 | 687.900 | 539.400 | 6188.19 | 1489.24 | 13.51 | 12.37 |
| | 2013 | 654.700 | 484.400 | 6807.43 | 1498.87 | 13.57 | 12.52 |
| | 2014 | 706.050 | 542.260 | 7593.88 | 1595.70 | 13.64 | 12.95 |
| | 2015 | 716.200 | 582.400 | 7924.65 | 1581.59 | 13.71 | 13.11 |
| | 2016 | 755.400 | 678.900 | 8000.34 | 1599.00 | 13.72 | 13.22 |
| | 2017 | 809.998 | 599.450 | 8123.22 | 1601.22 | 13.98 | 13.56 |
| | 2018 | 825.445 | 601.220 | 8256.77 | 1598.88 | 14.31 | 13.89 |
| $i$: China $j$: Pakistan | 2001 | 13.970 | 8.150 | 1041.64 | 490.04 | 12.72 | 1.48 |
| | 2002 | 17.995 | 12.420 | 1135.45 | 480.74 | 12.80 | 1.50 |
| | 2003 | 24.300 | 18.550 | 1273.64 | 543.59 | 12.88 | 1.53 |
| | 2004 | 30.610 | 24.660 | 1490.38 | 628.63 | 12.96 | 1.56 |
| | 2005 | 42.610 | 34.280 | 1731.13 | 690.85 | 13.04 | 1.59 |
| | 2006 | 52.460 | 42.390 | 2069.34 | 789.41 | 13.11 | 1.62 |
| | 2007 | 68.930 | 57.890 | 2651.26 | 870.63 | 13.18 | 1.64 |
| | 2008 | 70.570 | 60.510 | 3413.59 | 978.80 | 13.25 | 1.67 |
| | 2009 | 67.880 | 55.280 | 3748.93 | 949.12 | 13.31 | 1.70 |
| | 2010 | 86.680 | 69.370 | 4432.96 | 1018.87 | 13.38 | 1.74 |
| | 2011 | 105.570 | 84.390 | 5444.79 | 1194.33 | 13.44 | 1.77 |
| | 2012 | 124.130 | 92.750 | 6188.19 | 1290.36 | 13.51 | 1.79 |
| | 2013 | 142.150 | 110.190 | 6807.43 | 1299.12 | 13.57 | 1.82 |
| | 2014 | 160.060 | 132.480 | 7593.88 | 1334.15 | 13.64 | 1.85 |
| | 2015 | 189.300 | 164.500 | 7924.65 | 1428.99 | 13.71 | 1.89 |
| | 2016 | 219.000 | 189.000 | 8000.34 | 1521.80 | 13.72 | 1.99 |
| | 2017 | 225.340 | 210.990 | 8123.22 | 1788.90 | 13.98 | 1.97 |
| | 2018 | 279.900 | 235.790 | 8256.77 | 1899.90 | 14.31 | 1.89 |

*(Continued)*

**Table 2.** (Continued)

|  | Year | EAI$_{ijt}$ (Billion USD) | EXP$_{ijt}$ (Billion USD) | PGDP$_{it}$ (USD) | PGDP$_{jt}$ (USD) | P$_{it}$ (Billion) | P$_{jt}$ (Billion) |
|---|---|---|---|---|---|---|---|
| i: China j: Bangladesh | 2001 | 9.720 | 9.550 | 1041.64 | 356.12 | 12.72 | 1.32 |
|  | 2002 | 11.000 | 10.680 | 1135.45 | 354.30 | 12.80 | 1.34 |
|  | 2003 | 13.700 | 13.360 | 1273.64 | 380.28 | 12.88 | 1.37 |
|  | 2004 | 19.660 | 19.090 | 1490.38 | 407.99 | 12.96 | 1.39 |
|  | 2005 | 24.800 | 24.020 | 1731.13 | 428.75 | 13.04 | 1.41 |
|  | 2006 | 31.890 | 30.900 | 2069.34 | 434.84 | 13.11 | 1.42 |
|  | 2007 | 34.590 | 33.450 | 2651.26 | 475.25 | 13.18 | 1.44 |
|  | 2008 | 46.850 | 45.540 | 3413.59 | 546.85 | 13.25 | 1.45 |
|  | 2009 | 45.820 | 44.410 | 3748.93 | 607.76 | 13.31 | 1.47 |
|  | 2010 | 70.570 | 67.890 | 4432.96 | 674.93 | 13.38 | 1.49 |
|  | 2011 | 82.600 | 78.100 | 5444.79 | 735.00 | 13.44 | 1.50 |
|  | 2012 | 84.500 | 79.700 | 6188.19 | 747.34 | 13.51 | 1.55 |
|  | 2013 | 103.070 | 97.050 | 6807.43 | 829.25 | 13.57 | 1.57 |
|  | 2014 | 125.430 | 117.820 | 7593.88 | 1092.67 | 13.64 | 1.59 |
|  | 2015 | 147.070 | 139.010 | 7924.65 | 1211.70 | 13.71 | 1.61 |
|  | 2016 | 155.230 | 141.909 | 8000.34 | 1467.88 | 13.72 | 1.67 |
|  | 2017 | 159.990 | 145.078 | 8123.22 | 1541.11 | 13.98 | 1.78 |
|  | 2018 | 162.786 | 146.099 | 8256.77 | 1578.09 | 14.31 | 1.78 |
| i: China j: Afghanistan | 2001 | 0.174 | 0.172 | 1041.64 | 92.21 | 12.72 | 0.267 |
|  | 2002 | 0.199 | 0.199 | 1135.45 | 157.98 | 12.80 | 0.274 |
|  | 2003 | 0.271 | 0.265 | 1273.64 | 168.68 | 12.88 | 0.282 |
|  | 2004 | 0.579 | 0.569 | 1490.38 | 196.23 | 12.96 | 0.290 |
|  | 2005 | 0.527 | 0.512 | 1731.13 | 227.88 | 13.04 | 0.299 |
|  | 2006 | 1.006 | 1.004 | 2069.34 | 251.11 | 13.11 | 0.307 |
|  | 2007 | 1.718 | 1.694 | 2651.26 | 306.98 | 13.18 | 0.316 |
|  | 2008 | 1.543 | 1.516 | 3413.59 | 367.19 | 13.25 | 0.325 |
|  | 2009 | 2.148 | 2.135 | 3748.93 | 425.07 | 13.31 | 0.334 |
|  | 2010 | 1.789 | 1.752 | 4432.96 | 501.47 | 13.38 | 0.343 |
|  | 2011 | 2.344 | 2.300 | 5444.79 | 575.97 | 13.44 | 0.353 |
|  | 2012 | 4.692 | 4.640 | 6188.19 | 575.97 | 13.51 | 0.298 |
|  | 2013 | 3.378 | 3.282 | 6807.43 | 678.35 | 13.57 | 0.305 |
|  | 2014 | 4.109 | 3.393 | 7593.88 | 658.98 | 13.64 | 0.316 |
|  | 2015 | 3.760 | 3.640 | 7924.65 | 590.27 | 13.71 | 0.325 |
|  | 2016 | 3.980 | 3.780 | 8000.34 | 599.01 | 13.72 | 0.356 |
|  | 2017 | 4.009 | 3.889 | 8123.22 | 600.89 | 13.98 | 0.339 |
|  | 2018 | 4.145 | 3.990 | 8256.77 | 611.29 | 14.31 | 0.390 |
| i: China j: Bhutan | 2001 | 0.0162 | 0.0160 | 1041.64 | 774.89 | 12.72 | 0.0588 |
|  | 2002 | 0.0064 | 0.0062 | 1135.45 | 837.05 | 12.80 | 0.0606 |
|  | 2003 | 0.0198 | 0.0197 | 1273.64 | 978.51 | 12.88 | 0.0624 |
|  | 2004 | 0.0052 | 0.0035 | 1490.38 | 1074.58 | 12.96 | 0.0642 |
|  | 2005 | 0.0047 | 0.0047 | 1731.13 | 1242.04 | 13.04 | 0.0659 |
|  | 2006 | 0.0016 | 0.0016 | 2069.34 | 1330.52 | 13.11 | 0.0674 |
|  | 2007 | 0.0539 | 0.0539 | 2651.26 | 1736.97 | 13.18 | 0.0688 |
|  | 2008 | 0.0846 | 0.0846 | 3413.59 | 1792.91 | 13.25 | 0.0701 |
|  | 2009 | 0.0417 | 0.0412 | 3748.93 | 1772.10 | 13.31 | 0.0713 |
|  | 2010 | 0.0160 | 0.0159 | 4432.96 | 2088.49 | 13.38 | 0.0725 |
|  | 2011 | 0.1746 | 0.1738 | 5444.79 | 2287.71 | 13.44 | 0.0738 |
|  | 2012 | 0.1562 | 0.1560 | 6188.19 | 2398.91 | 13.51 | 0.0741 |
|  | 2013 | 0.1741 | 0.1740 | 6807.43 | 2498.39 | 13.57 | 0.0753 |
|  | 2014 | 0.1122 | 0.1112 | 7593.88 | 2380.91 | 13.64 | 0.0765 |
|  | 2015 | 0.1531 | 0.1520 | 7924.65 | 2532.45 | 13.71 | 0.0774 |
|  | 2016 | 0.1677 | 0.1677 | 8000.34 | 2667.89 | 13.72 | 0.0788 |
|  | 2017 | 0.1998 | 0.1690 | 8123.22 | 2776.90 | 13.98 | 0.0791 |
|  | 2018 | 0.2319 | 0.1709 | 8256.77 | 2890.00 | 14.31 | 0.0801 |

*(Continued)*

**Table 2.** (Continued)

| | Year | EAI$_{ijt}$ (Billion USD) | EXP$_{ijt}$ (Billion USD) | PGDP$_{it}$ (USD) | PGDP$_{jt}$ (USD) | P$_{it}$ (Billion) | P$_{jt}$ (Billion) |
|---|---|---|---|---|---|---|---|
| i: China j: Sri Lanka | 2001 | 3.967 | 3.866 | 1041.64 | 837.70 | 12.72 | 0.187 |
| | 2002 | 3.510 | 3.367 | 1135.45 | 903.90 | 12.80 | 0.189 |
| | 2003 | 5.240 | 5.040 | 1273.64 | 984.81 | 12.88 | 0.191 |
| | 2004 | 7.170 | 6.940 | 1490.38 | 1063.16 | 12.96 | 0.194 |
| | 2005 | 9.760 | 9.390 | 1731.13 | 1242.40 | 13.04 | 0.196 |
| | 2006 | 11.410 | 11.060 | 2069.34 | 1423.48 | 13.11 | 0.198 |
| | 2007 | 14.320 | 13.840 | 2651.26 | 1614.41 | 13.18 | 0.200 |
| | 2008 | 16.900 | 16.300 | 3413.59 | 2013.91 | 13.25 | 0.202 |
| | 2009 | 16.390 | 15.680 | 3748.93 | 2057.11 | 13.31 | 0.204 |
| | 2010 | 20.970 | 19.940 | 4432.96 | 2400.02 | 13.38 | 0.206 |
| | 2011 | 31.410 | 29.880 | 5444.79 | 2835.41 | 13.44 | 0.208 |
| | 2012 | 31.630 | 30.010 | 6188.19 | 2923.13 | 13.51 | 0.203 |
| | 2013 | 36.190 | 34.360 | 6807.43 | 3279.89 | 13.57 | 0.204 |
| | 2014 | 40.410 | 37.920 | 7593.88 | 3631.05 | 13.64 | 0.206 |
| | 2015 | 40.300 | 37.300 | 7924.65 | 3926.17 | 13.71 | 0.209 |
| | 2016 | 41.009 | 37.890 | 8000.34 | 4009.31 | 13.72 | 0.208 |
| | 2017 | 41.890 | 37.990 | 8123.22 | 4109.90 | 13.98 | 0.210 |
| | 2018 | 42.009 | 38.009 | 8256.77 | 4289.99 | 14.31 | 0.221 |
| i: China j: Maldives | 2001 | 0.0220 | 0.0210 | 1041.64 | 2838.07 | 12.72 | 0.00277 |
| | 2002 | 0.0298 | 0.0296 | 1135.45 | 2885.49 | 12.80 | 0.00282 |
| | 2003 | 0.0335 | 0.0334 | 1273.64 | 3251.62 | 12.88 | 0.00286 |
| | 2004 | 0.0809 | 0.0791 | 1490.38 | 3638.57 | 12.96 | 0.00291 |
| | 2005 | 0.1696 | 0.1693 | 1731.13 | 3361.59 | 13.04 | 0.00295 |
| | 2006 | 0.1600 | 0.1540 | 2069.34 | 4353.01 | 13.11 | 0.00299 |
| | 2007 | 0.2500 | 0.2470 | 2651.26 | 5079.99 | 13.18 | 0.00303 |
| | 2008 | 0.3290 | 0.3150 | 3413.59 | 6149.01 | 13.25 | 0.00307 |
| | 2009 | 0.4080 | 0.4060 | 3748.93 | 6229.67 | 13.31 | 0.00311 |
| | 2010 | 0.6350 | 0.6340 | 4432.96 | 6570.43 | 13.38 | 0.00315 |
| | 2011 | 0.9730 | 0.9710 | 5444.79 | 6405.05 | 13.44 | 0.00320 |
| | 2012 | 0.7660 | 0.7640 | 6188.19 | 6566.65 | 13.51 | 0.00338 |
| | 2013 | 0.9780 | 0.9740 | 6807.43 | 6665.77 | 13.57 | 0.00345 |
| | 2014 | 1.0430 | 1.0390 | 7593.88 | 8483.81 | 13.64 | 0.00357 |
| | 2015 | 1.0920 | 1.0400 | 7924.65 | 7681.08 | 13.71 | 0.00409 |
| | 2016 | 1.1098 | 1.0448 | 8000.34 | 7767.89 | 13.72 | 0.00412 |
| | 2017 | 1.0989 | 1.0489 | 8123.22 | 7890.90 | 13.98 | 0.00467 |
| | 2018 | 1.0998 | 1.0509 | 8256.77 | 7998.12 | 14.31 | 0.00489 |

After analyzing the Table 5, we could obtain the following conclusions:

i. For both the exports model and bilateral trade model, $\gamma \neq 0$ means non-efficiency trade's influence on trade efficiency in two models are 84.33% and 99.99%, respectively. The exports model's $\gamma$ value is larger than the bilateral trade model, which means exports trade resistance from China to Nepal is larger than the bilateral trade resistance.

ii. For both the exports model and bilateral trade model, $\mu \neq 0$ shows that the non-efficiency factors existed in the trade processes, the SFGM is suitable to describe the trade between China and Nepal.

iii. For the exports model and bilateral trade model, $\mu \neq 0$ shows that the time-varying SFGMs are suitable to describe the trade between China and Nepal. For the bilateral trade model, $\eta = 0.057$ shows the bilateral non-efficiency factor decreasing at a rate of 0.057 with time

**Table 3. Other initial data of the eight South Asian countries.**

|  | $D_{ij}$ (kilometer) | $B_{ij}$ | $LAN_{ij}$ | $LANG_{ij}$ | $ETA_{ij}$ |
|---|---|---|---|---|---|
| *i*: China, *j*: Nepal | 1976.1 | 1 | 0 | 1 | 0 |
| *i*: China, *j*: India | 2620.7 | 1 | 0 | 1 | 0 |
| *i*: China, *j*: Pakistan | 2802.8 | 1 | 0 | 1 | 1 |
| *i*: China, *j*: Bangladesh | 1874.6 | 0 | 0 | 1 | 0 |
| *i*: China, *j*: Afghanistan | 3129.0 | 1 | 0 | 1 | 0 |
| *i*: China, *j*: Bhutan | 1644.0 | 1 | 0 | 1 | 0 |
| *i*: China, *j*: Sri Lanka | 4043.0 | 0 | 0 | 1 | 0 |
| *i*: China, *j*: Maldives | 4711.0 | 0 | 0 | 1 | 0 |

increasing, and the bilateral trade increasing at a rate of 0.057 with time increasing; For the exports model, $\eta = -0.004$ means the exports non-efficiency factor increasing at a rate of 0.004 with time increasing, exports trade decreasing at a rate of 0.057 with time increasing.

iv. For both the exports model and bilateral trade model, elasticity of per capita GDP in China is less than the value of the other seven trade countries, shows that the economic size of the trade countries has a great influence on China's exports and bilateral trade. It is necessary to evaluate the seven trade countries' economic development in the future if China wants more trades among the seven countries.

v. The coefficients of the trade countries' population are negative in the two models, indicates that if the trade countries have small number of population, they have smaller domestic markets, and they have smaller imports from China as well as bilateral trades. On the contrary, the coefficients of China's population are negative in the two models; shows that the larger population constitutes a larger domestic market and larger imports demand form the trade countries. We thought that China's labor force advantages have changed into the trade advantages.

vi. The coefficients of the distance between two trade countries are negative in the two models. When the distance between two trading countries increased 1%, the exports trade value decreased 3.009%, and the bilateral trade value decreased 0.059%. Longer distance means larger transportation costs, which limits the development of trades.

vii. The coefficients of B and $LAN_{ij}$ are negative in the exports model. Most of the countries sharing a common border with China are located in inner Asia-Europe Continent, the land transportation corridors development between two bordering countries are limited by the geographical environment, e.g., the mountains. Compared with the landlocked countries, China has both territorial land and territorial sea, the shipping provides a greater convenience for China's exports trade.

viii. The coefficients of $LANG_{ij}$ are positive in the two models, English is the commonly used language among China and other counties while trading, it is helpful to improve the trade development if two trading countries share one common language.

**Table 4. The model testing results.**

|  | $H_0$ | $\ln(H_0)$ | $\ln(H_1)$ | LR statistics value | Critical value (1%) | Conclusion |
|---|---|---|---|---|---|---|
| Eq (6) | $\gamma = \mu = \eta = 0$ | -173.2444 | -299.4199 | 252.3511 | 11.34 | Refuse |
|  | $\eta = 0$ | -299.4199 | -391.4562 | 183.8726 | 6.63 | Refuse |
| Eq (7) | $\gamma = \mu = \eta = 0$ | -302.8503 | -342.9575 | 80.2145 | 9.21 | Refuse |
|  | $\eta = 0$ | -262.7431 | -272.3425 | 19.1988 | 6.63 | Refuse |

**Table 5. The basic regression analysis and model checking results based on SFGM.**

| Independent variable | Exports model | | | Bilateral trade model | | |
|---|---|---|---|---|---|---|
| | Coefficient | T statistic | P value | Coefficient | T statistic | P value |
| Constant term | 8.1740 | 5.7900 | 0.0000 | 24.5357 | 24.8255 | 0.0000 |
| $\ln PGDP_i$ | 0.2052 | 3.6128 | 0.0026 | -0.0711 | -0.6141 | 0.4635 |
| $\ln PGDP_j$ | 2.0500 | 13.0621 | 0.0000 | 0.1449 | 0.6506 | 0.4356 |
| $\ln P_i$ | 7.2513 | 2.4249 | 0.0039 | 0.0329 | 0.0347 | 0.9268 |
| $\ln P_j$ | -2.2414 | -15.9151 | 0.0000 | -2.0240 | -7.3138 | 0.0000 |
| $\ln D_{ij}$ | -3.0095 | -4.6072 | 0.0022 | -0.0596 | -0.6811 | 0.0001 |
| $B_{ij}$ | 0.5208 | 0.8756 | 0.4852 | 0.2101 | 0.2490 | 0.5486 |
| $LAN_{ij}$ | -0.1630 | -0.1650 | 0.8365 | -1.7590 | -10.3410 | 0.0000 |
| $LANG_{ij}$ | 5.0837 | 0.9458 | 0.3256 | 0.0127 | 0.0128 | 0.9325 |
| $FTA_{ij}$ | 1.2898 | 1.5898 | 0.1625 | 0.1515 | 0.1528 | 0.8735 |
| $\delta^2$ | 2.2449 | 6.8936 | 0.0001 | 21.3774 | 21.8081 | 0.0000 |
| $\gamma$ | 0.9999 | 432.9989 | 0.0000 | 0.8433 | 17302.4610 | 0.0000 |
| $\mu$ | -2.5120 | -4.8310 | 0.0019 | 0.0850 | 0.0850 | 0.9023 |
| $\eta$ | -0.0040 | -0.1850 | 0.8269 | 0.0570 | 0.0670 | 0.9125 |
| Log likelihood | -173.2440 | —— | —— | -262.7431 | —— | —— |
| LR Statistics | 252.3511 | —— | —— | 80.2145 | —— | —— |

ix. The coefficients of $FTA_{ij}$ are positive in the two models, which means the free trade agreement is critical to reduce bilateral trade barriers and trade non-efficiency factors.

The results in Table 6 show that larger efficiency value, larger EE and BTE, and smaller trade potential. Otherwise, smaller efficiency value, smaller EE and BTE, means larger trade potential. The BTE between China and Nepal increases when time changes, the EE from China to Nepal remains constant changing during the 18 years. The changing range of BTE is 0.002–0.05; the changing range of EE from China to Nepal is over 0.1, larger than the BTE. The BTE and EE ranking among the eight South Asian countries are ranking fifth and fourth during the 18 years (BTE ranking from 1 to 8: India, Pakistan, Bangladesh, Sri Lanka, Nepal, Afghanistan, Maldives, Bhutan; EE ranking from 1 to 8: India, Bangladesh, Pakistan, Nepal, Afghanistan, Bhutan, Sri Lanka, Maldives). The increasing BTE shows that the trade between China and Nepal has improved a lot; the trade potential has a lot of room to improve. Larger EE means the smaller exports trade potential, increasingly saturated exports markets from China to Nepal, some measures must be applied to improve China's trade surplus.

**Table 6. Exports efficiency (EE) and bilateral trade efficiency (BTE) results between China and Nepal based on SFGM.**

| Year | EE | EE Rank | BTE | BTE Rank | Year | EE | EE Rank | BTE | BTE Rank |
|---|---|---|---|---|---|---|---|---|---|
| 2001 | 0.1324 | 4 | 0.0025 | 5 | 2010 | 0.0863 | 4 | 0.0124 | 5 |
| 2002 | 0.0744 | 4 | 0.0020 | 5 | 2011 | 0.1313 | 4 | 0.0179 | 5 |
| 2003 | 0.1053 | 4 | 0.0021 | 5 | 2012 | 0.1280 | 4 | 0.0322 | 5 |
| 2004 | 0.0837 | 4 | 0.0031 | 5 | 2013 | 0.1850 | 4 | 0.0338 | 5 |
| 2005 | 0.1035 | 4 | 0.0032 | 5 | 2014 | 0.1514 | 4 | 0.0378 | 5 |
| 2006 | 0.0883 | 4 | 0.0047 | 5 | 2015 | 0.1986 | 4 | 0.0434 | 5 |
| 2007 | 0.1379 | 4 | 0.0063 | 5 | 2016 | 0.2088 | 4 | 0.0452 | 5 |
| 2008 | 0.0688 | 4 | 0.0065 | 5 | 2017 | 0.2134 | 4 | 0.0466 | 5 |
| 2009 | 0.0959 | 4 | 0.0064 | 5 | 2018 | 0.2451 | 4 | 0.0478 | 5 |

**Table 7. Trade potential measurement results between China and Nepal (Billion USD).**

| Year | Exports trade potential from China to Nepal | | | | Bilateral trade potential | | | |
|------|--------|---------|---------|---------|--------|---------|---------|---------|
|      | EXP | ETP | EETP | IETP | EAI | BTP | EBTP | IBTP |
| 2001 | 1.486 | 11.226 | 11.362 | 1.20% | 1.532 | 611.131 | 873.801 | 42.98% |
| 2002 | 1.051 | 14.136 | 18.847 | 33.33% | 1.103 | 556.352 | 759.760 | 36.56% |
| 2003 | 1.220 | 11.590 | 11.146 | -3.83% | 1.273 | 607.900 | 796.939 | 31.10% |
| 2004 | 1.632 | 19.489 | 24.563 | 26.03% | 1.715 | 561.220 | 703.717 | 25.39% |
| 2005 | 1.879 | 18.162 | 16.521 | -9.03% | 1.964 | 621.060 | 743.909 | 19.78% |
| 2006 | 2.598 | 29.419 | 32.576 | 10.73% | 2.680 | 567.938 | 615.128 | 8.31% |
| 2007 | 3.860 | 27.982 | 24.368 | -12.92% | 4.000 | 631.958 | 698.734 | 10.57% |
| 2008 | 3.750 | 54.532 | 61.257 | 12.33% | 3.810 | 586.660 | 622.423 | 6.10% |
| 2009 | 4.090 | 42.652 | 35.940 | -15.74% | 4.140 | 643.052 | 662.756 | 3.06% |
| 2010 | 7.322 | 84.851 | 91.225 | 7.51% | 7.437 | 600.222 | 594.806 | -0.90% |
| 2011 | 11.810 | 89.917 | 72.541 | -19.32% | 11.950 | 667.626 | 638.855 | -4.31% |
| 2012 | 19.680 | 153.796 | 208.593 | 35.63% | 19.980 | 620.408 | 733.035 | 18.15% |
| 2013 | 22.100 | 119.461 | 123.302 | 3.22% | 22.540 | 666.582 | 769.951 | 15.51% |
| 2014 | 22.830 | 150.808 | 200.647 | 33.05% | 23.300 | 617.088 | 693.291 | 12.35% |
| 2015 | 27.890 | 140.424 | 142.450 | 1.44% | 28.660 | 659.878 | 724.848 | 9.85% |
| 2016 | 28.900 | 141.234 | 144.900 | 2.60% | 30.245 | 689.998 | 700.909 | 1.58% |
| 2017 | 31.008 | 150.900 | 157.770 | 4.55% | 32.335 | 699.908 | 711.789 | 1.70% |
| 2018 | 33.450 | 151.990 | 153.890 | 1.25% | 35.667 | 710.990 | 767.897 | 8.00% |

The ratio of actual trade volume (EXP and EAI) and efficiency (EE and BTE, presented in Table 6) represents the Exports Trade Potential (ETP) and Bilateral Trade Potential (BTP), respectively. We could obtain the Extended Exports Trade Potential (EETP) and Extended Bilateral Trade Potential (EBTP) by substituting the original data (presented in Tables 2 and 3) into the estimated random frontier gravity equations. The difference between ETP and EETP (BEP and EBTP) is, ETP (BEP) is influenced by the economic data of other seven countries, and EETP (EBTP) is the economic indicator which reflects the pure economic trade potential between China and Nepal. We subtracted 100% from the ratio of extended trade potential (EETP and EBTP) and trade potential (ETP and BTP), the results presented the Improved Exports Trade Potential (IETP) and Improved Bilateral Trade Potential (IBTP), respectively. All results were presented in Table 7.

The results in Table 7 show that the BTP is larger than ETP between China and Nepal, also we find that the average IBTP from 2001 is over 15%, but the average IETP from 2001 is just 6%, in 2003, 2005, 2007, 2009 and 2011 the IETP values are negative, which means the import trade potential from Nepal to China is huge, the focus of bilateral trade between China and Nepal may be changed, there are more goods may be exported from Nepal to China, and China may become trade deficit when trading with Nepal. Absolutely, the LKSARFT provides a convenient, safe, time-saving and cost-saving land transport corridor in the trades between China and Nepal.

Meanwhile, the LKSARFT developed a land route to connect the other South Asian countries. Nepal is located in the north of India, so the Kathmandu can act as the cargo transportation starting point to Bhutan, New Delhi, Kolkata and other South Asian cities. It is possible to connect the Kolkata and other South Asian countries such as Sri Lanka and Maldives through ocean shipping routes. The further development of the LKSARFT is helpful to improve the strategic spatial pattern of B&R, especially for the Maritime Silk Road.

## 4.2 The development bottlenecks and solutions for the LKSARFT

From the calculation and analysis results above, we can find that the development of LKSARFT is critical to the bilateral trade between China and Nepal, but after our field research we found that there are a lot of bottlenecks existed: (i) Infrastructure capacity limitation, e.g., capacity limitation in Golmud-Lhasa section of Qinghai-Tibet Railway, capacity limitation in Lhasa-Shigatse section of Qinghai-Tibet Railway. The Qinghai-Tibet Railway may not be good ones based on cost-benefit analysis considering the huge amount of investment, but it does play an important role in connecting the lagged hinterland areas and the developed eastern areas and hence are important in boosting the development of economic growth in western areas [43]. Furthermore, the terrible roadway operation condition from Geelong town, China to Kathmandu, Nepal is one of infrastructure capacity bottleneck. (ii) Operational and management defects, e.g., ambiguous responsibility of transport operators, imperfect subsidy and exit mechanism, the high unload ratio from Nepal to China and so on. (iii) Other development bottleneck, e.g., the chaotic political situation in Nepal, unbalanced bilateral trade, harsh natural environment, frequent natural disasters and so on. In order to ensure the development sustainability of LKSARFT, we now introduce these bottlenecks and the solutions in detail.

### 4.2.1 Infrastructure capacity limitation.

*(1) Capacity limitation in Golmud-Lhasa section of Qinghai-Tibet Railway*. The total railway line length of the Golmud-Lhasa section is 1142 kilometers, passing through 960 kilometers high altitude areas (with altitude over 4000 meters), and 550 kilometers permafrost areas. In order to ensure the passengers' safety, reduce the altitude sickness, the passenger trains must be operated during day time. The purple lines and red lines in Fig 2 show some parts

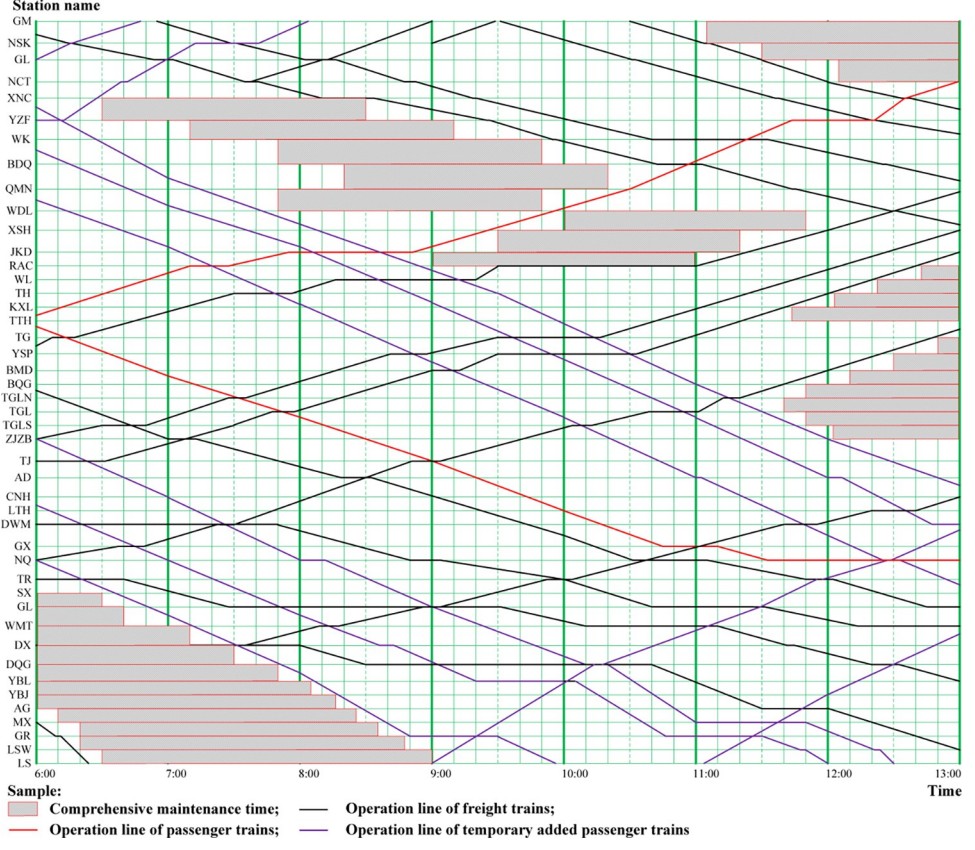

**Fig 2. The train operation diagram of Golmud-Lhasa section from 6:00 to 13:00 (December 15, 2016).**

of the passenger trains' operation lines in Golmud-Lhasa section, which must be intensive, continuous operation in the operation diagram. Furthermore, the comprehensive maintenance time for each section of the line is critical to make the operation safe, especially under harsh environment conditions. The comprehensive maintenance must be carried out during day time due to the low temperature during night in high altitude areas, therefore most of the cargo trains must be operated during night time. The Golmud-Lhasa railway line is single-track and non-electrified, the line designed capability is far lower than actual transportation demand capacity, because the cargo loaded on the freight trains are applied to support the construction of Tibet, especially for the Lhasa city. As a conclusion, the capacity shortage in Golmud-Lhasa section of Qinghai-Tibet Railway limits the development of LKSARFT, but China has already taken some measures to improve the Qinghai-Tibet railway line's capacity, for example, expanding some parts of the railway lines, building some new railway stations and rebuilding some parts of the single-track and non-electrified railway lines into double-tracks and electrified railway lines.

(2) *Capacity limitation in Lhasa-Shigatse section of Qinghai-Tibet Railway.* The Lhasa-Shigatse section of Qinghai-Tibet Railway is located in the southwest of the Qinghai-Tibet Plateau, which is quite close to Himalayas. With high altitude (over 3590 meters), deep ditch steep, geological complexity and fault, collapse, rock pile and other bad geologic sections, have great influence on the operation of the trains. When the sandstorm, snow damage freezing, landslide as well as the earthquake happened, most of the trains must be stopped in order to ensure the safety for both passengers and cargos. Meanwhile, the Lhasa-Shigatse line railway is single-track and electrified; the line design capability is lower than actual transportation demand capacity. The operation and organization is quite similar to Golmud-Lhasa section, passenger trains passed through during the day time and cargo trains during night time (showed in Fig 3), but the railway lines in Lhasa-Shigatse section needed longer comprehensive maintenance time when compared with railway lines in Golmud-Lhasa section (see Fig 2), because the natural environment of Lhasa-Shigatse section is harsher. We thought that how to make full use of the operation capacity of the railway operation diagram, reduce the comprehensive maintenance time by using advanced approaches, are the key issues for the managers and operators to follow with interest.

(3) *Terrible roadway operation conditions and tedious security check from Geelong town to Kathmandu.* Nepal is an agricultural based country, 80% of the population live on

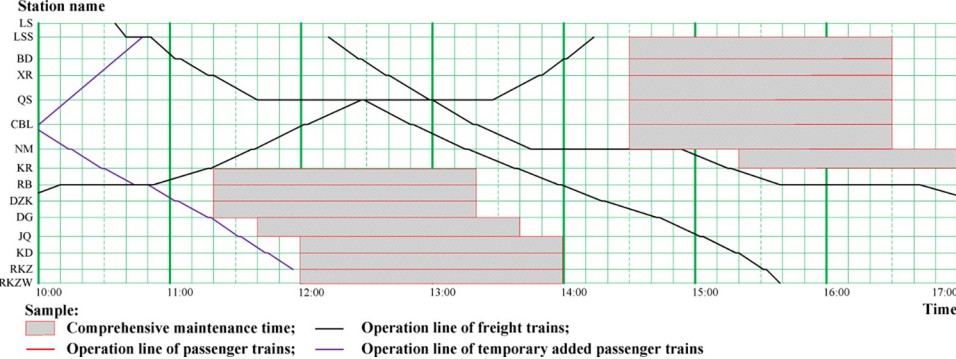

**Fig 3. The train operation diagram of Lhasa-Shigatse section from 10:00 to 17:00 (December 15, 2016).**

agriculture. It is one of the world's 48 least developed countries, with backward infrastructure such as roads, highways, railways and so on. The calculation data shows that each kilometer road serves 14,600 people, the density of roads per one hundred square kilometers is10.4, which is very low when compared the data with the other seven South Asian countries [18]. There is only one road connected Geelong town and Kathmandu. In April 2015, a strong earthquake destroyed the only land transport corridor connected China and Nepal. After that, an agreement on economic and technical cooperation between China and Nepal was signed by the two governments, China promised to provide help on the Nepal's infrastructure construction, brought better opportunities on development of the LKSARFT as well as the bilateral trades. The narrow and dangerous road is located at halfway of the mountains, with a lot of mud, water puddles and single lane. The landslides, debris flow, fog and so on happened frequently. Furthermore, we had tedious security check along the 184 kilometers travel distance, with total 12 times checks, half of the checks needed to show prosecutors the cargos loaded in the containers, which was a waste of time, extremely.

We highly recommended that the road from Geelong town to Kathmandu need to be rebuilt. The B&R provides better development opportunities; the Asian Infrastructure Development Bank (AIIB) may provide financial support. We think the key issues of the new project for the AIIB including: (i) Value Engineering (VE) researches for projects. VE is one of the proven management techniques in the construction industry, which is applied to improve the function and eliminate unnecessary costs, deal with the core competencies of projects, usually it is handled with secrecy [44–46]. (ii) Chinese element in the project. For example, Chinese project construction, management, standards and specifications, see [47]. (iii) Assessment of the project's life cycle. Life-cycle assessment (LCA) is an evaluation of the environmental load and energy consumption of goods and services during their total life cycle. LCA has been applied in assessing construction costs, and has become an important technique for improving construction sustainability [48–50].

**4.2.2 Operational and management bottlenecks.**

*(1) Ambiguous responsibility of transport operators*. There is no accurate goods delivery terms trade during the international railway combined transport processes. Although the General Rules for the Interpretation of International Trade Terms (2000 edition) is generic applied during the international cargo trade processes, but the international railway combined transport has its own characteristics, so it is frequent to mislead the foreign trade relations on the understanding of ambiguity. Furthermore, the unclear legal responsibility definition leads to unclear service specifications and diverse service standards.

*(2) Imperfect subsidy and exit mechanism*. The operation of LKSARFT is promoted by the government, not belongs to the complete market behavior. In order to support the LKSARFT trains' normalized operation, Lanzhou government will subside 10,000 CNY per container. Huge subsidies increase the government's financial burden, and there is no viable exit mechanism, which make the LKSARFT hard to continue. Huge subsidies also make the vicious competition among the operators when they collect the freight.

*(3) High train unload ratio from Nepal to China*. High train unload ratio from Nepal to China increased the LKSARFT trains' operation cost, one reason is the imported types of goods from Nepal are more lacking than the exported types of goods. Furthermore, the Nepalese businessmen have not better understood the newly operated LKSARFT trains.

## 4.3 Other development bottlenecks

**4.3.1 The chaotic political situation in Nepal.** Now there are nearly 70 parties in Nepal, the four main political parties including Nepalese Communist Party (Maoist), Nepal Congress Party, Joint Marxist-Leninist and Madisi Political Party determine the political situation in Nepal, the competition among the four main parties are fierce in recent years, which resulted in more and more difficult parliamentary elections and the formulation of new constitution anomalies, also affected the stabile relations between China and Nepal. Meanwhile, some international forces involved in the relationship between China and Nepal, e.g., India and the United States, a series of major changes in Nepal and its foreign policy will directly affect the development, security and stability of Tibet. Furthermore, Nepal is neighboring to Tibet, which provides geographical advantages for Dalai Lama to split the motherland illegally, e.g., the illegal crossing-border, illegal gathering etc., which seriously damaged the regional stability in Tibet, as well as the development of the LKSARFT.

**4.3.2 Unbalanced bilateral trade.** In 2015, bilateral trade between China and Nepal amounted to 28.66 billion USD, China's exports to Nepal amounted to 27.89 billion USD, while Nepal's exports to China were only 0.77 billion USD (see Table 2), the bilateral trades is unbalanced. Particularly, the trade between Tibet and Nepal occupies an important share, accounting for 87.91% of the total foreign trade of the Tibet Autonomous Region. Meanwhile, China mainly imports low value-added, low-technology, labor-intensive and resource-intensive products from Nepal, and exports technology-intensive goods to Nepal, and this trend expands continuously. The unbalanced trade policies and product structure are harmful to the development of LKSARFT.

**4.3.3 Harsh natural environment and frequent natural disasters.** Nepal and China's southwestern Tibet belong to the Himalayan plateau where the earthquakes happened frequently. April 25, 2015 and May 12, 2015, Nepal occurred 8.1 and 7.1 earthquake, respectively. 7903 people were killed and 16390 injured in Nepal, 26 people were killed and 856 injured in Tibet [51–53]. And this kind of natural disasters damages the roads and other infrastructures easily, e.g., the road from Geelong town to Kathmandu. The harsh natural environment and frequent natural disasters make the development of China-Nepal economic and trade, as well as the development of LKSARFT more difficult.

## 5. Conclusion and policy implications

The B&R provides good bilateral development opportunities for both China and Nepal, the LKSARFT acts as one of the important belts. In this paper we focused on two works: (i) studying the bilateral trade between China and Nepal, as well as the LKSARFT development prospects by applying SFGM, (ii) analyzing the development bottlenecks of LKSARFT according to the field research results.

Trades across borders are considered important in improving welfare of people in South Asian countries, we collected eight South Asian countries' basic data, and tried to use the initial data to study the prospects about LKSARFT based on SFGM, we want to formulate the trade potential of the two countries, find out the factors that promote or limit bilateral trade, evaluate the bilateral trade efficiency and the potential of bilateral trade. The GSFM analysis results showed that: (i) Exports trade resistance from China to Nepal is larger than the bilateral trade resistance. (ii) For the bilateral trade model, the bilateral non-efficiency factor decreasing at a rate of 0.057 with time increasing, bilateral trade increasing at a rate of 0.057 with time increasing. For the exports model, the exports non-efficiency factor increasing at a rate of 0.004 with time increasing, exports trade decreasing at a rate of 0.057 with time increasing. (iii) The economic size of the trade countries has a great influence on China's exports and bilateral trade,

the trade countries with small number of population, also have smaller domestic markets, and have smaller imports from China and bilateral trades. The larger population constitutes a larger domestic market and larger imports demands form the trade countries. (iv) Longer distance means larger transportation costs, which limits the development of trades. (v) English is the commonly used language among China and other counties while trading, it is helpful to improve the trade development if two trading countries share one common language. The free trade agreement is critical to reduce bilateral trade barriers and trade non-efficiency factors. (vi) The BTE between China and Nepal increases when time changes, the EE from China to Nepal remains constant changing during the 18 years. The changing range of BTE is 0.002–0.05; the changing range of EE from China to Nepal is over 0.1, larger than the BTE. The BTE and EE ranking among the eight South Asian countries are ranking fifth and fourth during the 18 years. (vii) The increasing BTE shows that the trade between China and Nepal has improved a lot; the trade potential has a lot of room to improve. Larger EE means the smaller exports trade potential, increasingly saturated exports markets from China to Nepal, some measures must be applied to improve China's trade surplus. The BTP is larger than ETP between China and Nepal. The import trade potential from Nepal to China is huge, the focus of bilateral trade between China and Nepal may be changed, there are more goods may be exported from Nepal to China, and China may become trade deficit when trading with Nepal.

The bottleneck for the development of LKSARFT including: capacity limitation in Golmud-Lhasa section of Qinghai-Tibet Railway, capacity limitation in Lhasa-Shigatse section of Qinghai-Tibet Railway, terrible roadway operation conditions and tedious security check from Geelong town to Kathmandu, ambiguous responsibility of transport operators, imperfect subsidy and exit mechanism, the high unload ratio from Nepal to China, the chaotic political situation in Nepal, unbalanced bilateral trade, harsh natural environment and frequent natural disasters and so on. We gave some of our advisement for better LKSARFT, such as expanded rebuilt some parts of the railway lines, built some new railway stations and rebuilt the single-track and non-electrified railway lines into double-tracks and electrified railway lines in Golmud-Lhasa section of Qinghai-Tibet Railway; reduced the comprehensive maintenance time by using advanced approaches in Lhasa-Shigatse section; rebuilt the road from Geelong town to Kathmandu etc.

In line with these findings, we give policy directions to boost bilateral trade efficiency and tap the potential of bilateral trade between the two countries: (i) Export goods would be stimulated by formulating and implementing macro-economic policies aimed at increasing the economic size of China. (ii) The establishment of a robust and comprehensive integrated process for foreign exports would facilitate the process of foreign exports, consequently reducing China's export trade impediments with Nepal. (iii) In order to mitigate transportation costs, China should establish and implement tax incentive policies for cross-border trade in encouraged sectors. (iv) Both countries should promote the teaching of each other's languages in their respective education systems to enhance communication and cooperation, and facilitate cross-cultural understanding, thereby promoting trade development between the two nations. (iv) To rectify China's trade surplus, it is imperative for the nation to calibrate its fiscal policies such that there is a surge in demand, thereby inducing an increase in import volume.

Some basic problems, such as operational and management bottlenecks still remained unresolved. How to solve the questions mentioned above, are the further study works; furthermore, distance is the key variable in this paper, and it reflects the transportation cost between two countries and inversely proportional to trade, and the traffic routes and transport modes between China and South Asian Countries are different, so next we will try to improve our work using the distance (between ports, main trading city or capital) multiplied by freight rate (ocean freight, road freight or rail freight) to analyze the bilateral trade between two countries;

LKSARFT may reduce the transportation cost between China and Nepal, and even promote trade between China and other South Asian Countries, the new transport lines of LKSARFT are under construction, so next we can also introduce new distance reduced by LKSARFT in the SFGM to predict the trade potential under LKSARFT. Moreover, in order to refine the study, we will discuss the significance of each variable and examine impact of factors that affect trading potential in the future.

## Author Contributions

**Writing – original draft:** Fei Tian.

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
