## [Decision Letter · Decision Letter 0]

27 Feb 2023

PONE-D-23-03918Bilateral trade potential analysis of a rail-road freight train linking developing countries: A Stochastic Frontier Gravity Model approachPLOS ONE

Dear Dr. Tian,

Thank you for submitting your manuscript to PLOS ONE. After careful consideration, we feel that it has merit but does not fully meet PLOS ONE’s publication criteria as it currently stands. Therefore, we invite you to submit a revised version of the manuscript that addresses the points raised during the review process.

We look forward to receiving your revised manuscript.

Kind regards,

Carlos Alberto Zúniga-González, Ph.D

Academic Editor

PLOS ONE

Journal Requirements:

"This research was supported by the National Natural Science Foundation of China (71173177). The authors would like to thank the anonymous referees for their valuable comments and suggestions."

"This research was supported by the National Natural Science Foundation of China (71173177). The authors would like to thank the anonymous referees for their valuable comments and suggestions."

"This research was supported by the National Natural Science Foundation of China (71173177). The authors would like to thank the anonymous referees for their valuable comments and suggestions."

"The authors state that there is no competing interest."

7. PLOS requires an ORCID iD for the corresponding author in Editorial Manager on papers submitted after December 6th, 2016. Please ensure that you have an ORCID iD and that it is validated in Editorial Manager. To do this, go to ‘Update my Information’ (in the upper left-hand corner of the main menu), and click on the Fetch/Validate link next to the ORCID field. This will take you to the ORCID site and allow you to create a new iD or authenticate a pre-existing iD in Editorial Manager. Please see the following video for instructions on linking an ORCID iD to your Editorial Manager account: https://www.youtube.com/watch?v=_xcclfuvtxQ

8. We note that Figure 2 in your submission contain [map/satellite] images which may be copyrighted. All PLOS content is published under the Creative Commons Attribution License (CC BY 4.0), which means that the manuscript, images, and Supporting Information files will be freely available online, and any third party is permitted to access, download, copy, distribute, and use these materials in any way, even commercially, with proper attribution. For these reasons, we cannot publish previously copyrighted maps or satellite images created using proprietary data, such as Google software (Google Maps, Street View, and Earth). For more information, see our copyright guidelines: http://journals.plos.org/plosone/s/licenses-and-copyright.

a. You may seek permission from the original copyright holder of Figure 2 [#] to publish the content specifically under the CC BY 4.0 license.  

Additional Editor Comments:

Dear author I have reviewed your manuscript where you focused the stochastic frontier gravity model is applied to analyze the trade potential between China and Nepal and the prospects of Lanzhou-Kathmandu South

Asian rail-road freight trains (LKSARFT), I invite you to make the improvement according reviewers' observations. Regarding to the observation reviewer in the point 4 I would like to suggest some references that you may be to change or add. In the first you can find stochastic Frontier models, and the second is a application of stochastic frontier model.

[1] Dios-Palomares et al. (2015) Analysis of the efficiency of farming systems in Latin America and the Caribbean considering environmental issues Revista Cientifica de la Facultad de Ciencias Veterinarias de la Universidad del Zulia, 2015, 25(1), pp. 43–50 (In Scupus)

[2] Zuniga-Gonzalez, C.A (2011). Technical efficiency of organic fertilizer in small farms of Nicaragua: 1998-2005. AFRICAN JOURNAL OF BUSINESS MANAGEMENT Volume5 Issue3 Page967-973. https://www.webofscience.com/wos/woscc/full-record/WOS:000290682800034 (In WOS)

The data and information of this manuscript is relevant for publication in this journal, but you need to make the improvement by reviewers indicated.

Reviewers' comments:

Reviewer's Responses to Questions

**Comments to the Author**

1. Is the manuscript technically sound, and do the data support the conclusions?

Reviewer #1: Yes

Reviewer #2: Partly

2. Has the statistical analysis been performed appropriately and rigorously? 

Reviewer #1: Yes

Reviewer #2: Yes

3. Have the authors made all data underlying the findings in their manuscript fully available?

Reviewer #1: Yes

Reviewer #2: Yes

4. Is the manuscript presented in an intelligible fashion and written in standard English?

Reviewer #1: Yes

Reviewer #2: Yes

5. Review Comments to the Author

Reviewer #1: This paper is written well and explained clearly the Bilateral trade potential analysis of a rail-road freight train linking developing countries: A Stochastic Frontier Gravity Model approach. However, I have few comments that would help improve the quality of the paper.

1. It would be better if the author reports the P-value of Table 4 or indicate the statistical significance level using appropriate indicator.

2. The authors need to discuss more policy implications of there study.

3. The authors reveal within the manuscript that the data set consist Bangladesh, Pakistan, India, Bhutan, Sri Lanka, Nepal Afghanistan, & Maldives. But the authors give much emphasis on Nepal only. I understand this, the discussion also must mention other countries involve.

4. The literature reviewed are inadequate, a lot of recent related studies are exempted from the review studies such as: Abdullahi et. al. (2022). Examining the determinants and efficiency of China’s agricultural exports using a stochastic frontier gravity model. Abdullahi, Aluko & Huo. (2021). Determinants, efficiency and potential of agri-food exports from Nigeria to the EU: Evidence from the stochastic frontier gravity model. Abdullahi et al. (2021). Determinants and Potential of Agri-Food Trade Using the Stochastic Frontier Gravity Model: Empirical Evidence From Nigeria. Jiang & Zhang (2021). Trade Sustainability and Efficiency under the Belt and Road Initiative: A Stochastic Frontier Analysis of China’s Trade Potential at Industry Level.

Reviewer #2: 1. Title: The manuscript title is general and it is recommended to make it specific.

2. Abstract: This section has been descriptively written. It is suggested to include the findings of the study in quantitate form here. In addition, policy implications should be included concisely at the end of this section.

3. The manuscript is not properly organized in terms of (a). Introduction (b). Literature Review (c). Methodology of the study (d). Results and discussion (e). Conclusion and policy Implications.

4. The novelty of the study is not properly outlined. It is recommended to include the novelty of the study at the end of the introduction.

5. It is recommended to include a separate section on “Literature Review”. It is recommended to include in this section a review of previous studies directly or indirectly related to this study. Furthermore, it is recommended to find out the research gap that this study fills.

6. It is recommended that a separate section on “methodology” should include (a). a table regarding “description of the variables” used in this study. (b). It is recommended to discuss the significance of each variable included in this study/model (c). Methods for measuring trading potential between trading partners should be included. (d). It is recommended to include the econometric model that examines the impact of factors including BRI that affect trading potential.

7. Referred to Table 1, it is recommended to include descriptive statistics for the data series rather than including the full data.

8. It is recommended to justify why the author used the SFGM model instead of other econometric models.

9. It is recommended to correlate your study with previous studies conducted. In addition, it is recommended to prescribe the policy implications based on the study results in a separate paragraph after the study conclusions.

10. It is recommended to enrich the manuscript with a literature review.

6. PLOS authors have the option to publish the peer review history of their article (what does this mean?). If published, this will include your full peer review and any attached files.

Reviewer #1: **Yes: **Nazir Muhammad Abdullahi

Reviewer #2: No

---

## [Author Response · Author response to Decision Letter 0]

1 Apr 2023

Additional Editor Comments:

Dear author I have reviewed your manuscript where you focused the stochastic frontier gravity model is applied to analyze the trade potential between China and Nepal and the prospects of Lanzhou-Kathmandu South

Asian rail-road freight trains (LKSARFT), I invite you to make the improvement according reviewers' observations. Regarding to the observation reviewer in the point 4 I would like to suggest some references that you may be to change or add. In the first you can find stochastic Frontier models, and the second is a application of stochastic frontier model.

[1] Dios-Palomares et al. (2015) Analysis of the efficiency of farming systems in Latin America and the Caribbean considering environmental issues Revista Cientifica de la Facultad de Ciencias Veterinarias de la Universidad del Zulia, 2015, 25(1), pp. 43–50 (In Scupus)

[2] Zuniga-Gonzalez, C.A (2011). Technical efficienc stochastic y of organic fertilizer in small farms of Nicaragua: 1998-2005. AFRICAN JOURNAL OF BUSINESS MANAGEMENT Volume5 Issue3 Page967-973. https://www.webofscience.com/wos/woscc/full-record/WOS:000290682800034 (In WOS)

Author’s respond: We have added more descriptions about previous works in the literature review.

The SFGM is the Integration of Gravity Model and Stochastic Frontier Production Function Model which was formally introduced by [26] to address the inherent bias of the conventional gravity model of trade and to estimate potential trade flows. [53] applied frontier production function analysis to small farms in Nicaragua during 1998-2005, the results showed an acceptable average of technical efficiency which the makers of public policy in Nicaragua a must consider for the future. [40] focused on examining Philippines’s exports efficiency and potential based on trading partner’s characteristics using SFGM, unlike the usual measure of gravity model using OLS that measure potential from the mean. [12] provided a survey about environmental efficiency measure issue. the paper deals with different ways for including environmental variables, which offered several perspectives for measuring efficiency with frontier methods. [45] estimated the potential and efficiency of trade flows between China and countries along the Maritime Silk Road from 2005 to 2013 by using the SFGM. Furthermore, [17] defined the extent of the Belt and Road in terms of geographical boundaries, justified the application of the SFGM to the foreign direct investment (FDI) analysis, and constructed a frontier regression model to assess the China’s outward FDI efficiency. 

[12] Dios-Palomares, R., Alcaide-López-de-Pablo, D., 2015. Analysis of the Efficiency of Farming Systems in Latin America and the Caribbean Considering Environmental Issues. Revista Científica de Veterinaria 25(1), 43–50.

[53] Zuniga-Gonzalez, C.A., 2011. Technical efficiency of organic fertilizer in small farms of Nicaragua: 1998-2005. African Journal of Business Management 5(3), 967-973.

The data and information of this manuscript is relevant for publication in this journal, but you need to make the improvement by reviewers indicated.

Reviewers' comments:

Reviewer's Responses to Questions

Comments to the Author

1. Is the manuscript technically sound, and do the data support the conclusions?

Reviewer #1: Yes

Reviewer #2: Partly

2. Has the statistical analysis been performed appropriately and rigorously?

Reviewer #1: Yes

Reviewer #2: Yes

3. Have the authors made all data underlying the findings in their manuscript fully available?

Reviewer #1: Yes

Reviewer #2: Yes

4. Is the manuscript presented in an intelligible fashion and written in standard English?

Reviewer #1: Yes

Reviewer #2: Yes

5. Review Comments to the Author

Reviewer #1: This paper is written well and explained clearly the Bilateral trade potential analysis of a rail-road freight train linking developing countries: A Stochastic Frontier Gravity Model approach. However, I have few comments that would help improve the quality of the paper.

1. It would be better if the author reports the P-value of Table 4 or indicate the statistical significance level using appropriate indicator.

Author’s respond: We have conducted statistical analysis on the P-value. Please see the additional content in Table 4.

Table 4

The basic regression analysis and model checking results based on SFGM

Independent variable Exports model Bilateral trade model

 Coefficient T statistic P value Coefficient T statistic P value

Constant term 8.1740 5.7900 0.0000 24.5357 24.8255 0.0000

0.2052 3.6128 0.0026 -0.0711 -0.6141 0.4635

2.0500 13.0621 0.0000 0.1449 0.6506 0.4356

7.2513 2.4249 0.0039 0.0329 0.0347 0.9268

-2.2414 -15.9151 0.0000 -2.0240 -7.3138 0.0000

-3.0095 -4.6072 0.0022 -0.0596 -0.6811 0.0001

0.5208 0.8756 0.4852 0.2101 0.2490 0.5486

-0.1630 -0.1650 0.8365 -1.7590 -10.3410 0.0000

5.0837 0.9458 0.3256 0.0127 0.0128 0.9325

1.2898 1.5898 0.1625 0.1515 0.1528 0.8735

2.2449 6.8936 0.0001 21.3774 21.8081 0.0000

0.9999 432.9989 0.0000 0.8433 17302.4610 0.0000

-2.5120 -4.8310 0.0019 0.0850 0.0850 0.9023

-0.0040 -0.1850 0.8269 0.0570 0.0670 0.9125

Log likelihood -173.2440 —— —— -262.7431 —— ——

LR Statistics 252.3511 —— —— 80.2145 —— ——

2. The authors need to discuss more policy implications of there study.

Author’s respond: We have prescribed the policy implications based on the study results in a separate paragraph after the study conclusions. Please see the Penultimate paragraph.

In line with these findings, we give policy directions to boost bilateral trade efficiency and tap the potential of bilateral trade between the two countries: (i) Export goods would be stimulated by formulating and implementing macro-economic policies aimed at increasing the economic size of China. (ii) The establishment of a robust and comprehensive integrated process for foreign exports would facilitate the process of foreign exports, consequently reducing China's export trade impediments with Nepal. (iii) In order to mitigate transportation costs, China should establish and implement tax incentive policies for cross-border trade in encouraged sectors. (iv) Both countries should promote the teaching of each other's languages in their respective education systems to enhance communication and cooperation, and facilitate cross-cultural understanding, thereby promoting trade development between the two nations. (iv) To rectify China's trade surplus, it is imperative for the nation to calibrate its fiscal policies such that there is a surge in demand, thereby inducing an increase in import volume.

3. The authors reveal within the manuscript that the data set consist Bangladesh, Pakistan, India, Bhutan, Sri Lanka, Nepal Afghanistan, & Maldives. But the authors give much emphasis on Nepal only. I understand this, the discussion also must mention other countries involve.

Author’s respond: We have added more descriptions about discussions pertaining to other countries in section 3 and section 4. Please see the red part.

Table 2

The needed initial data of the eight South Asian countries (from 2001 to 2018)

 Year 

(Billion USD) 

(Billion USD) 

(USD) 

(USD) 

(Billion) 

(Billion)

 : China

 : Nepal

2001 1.532 1.486 1041.64 240.47 12.72 0.250

 2002 1.103 1.051 1135.45 236.71 12.80 0.256

 2003 1.273 1.220 1273.64 242.14 12.88 0.261

 2004 1.715 1.632 1490.38 272.25 12.96 0.267

 2005 1.964 1.879 1731.13 298.01 13.04 0.273

 2006 2.680 2.598 2069.34 326.04 13.11 0.287

 2007 4.000 3.860 2651.26 362.22 13.18 0.284

 2008 3.810 3.750 3413.59 434.96 13.25 0.290

 2009 4.140 4.090 3748.93 438.29 13.31 0.294

 2010 7.437 7.322 4432.96 534.52 13.38 0.300

 2011 11.950 11.810 5444.79 619.45 13.44 0.305

 2012 19.980 19.680 6188.19 706.65 13.51 0.275

 2013 22.540 22.100 6807.43 694.10 13.57 0.278

 2014 23.300 22.830 7593.88 696.94 13.64 0.282

 2015 28.660 27.890 7924.65 732.30 13.71 0.285

 2016 30.245 28.900 8000.34 756.44 13.72 0.299

 2017 32.335 31.008 8123.22 788.00 13.98 0.310

 2018 35.667 33.450 8256.77 812.22 14.31 0.334

 : China

 : India

2001 35.960 18.960 1041.64 459.58 12.72 10.71

 2002 49.460 26.710 1135.45 480.21 12.80 10.89

 2003 75.950 33.430 1273.64 558.44 12.88 11.06

 2004 136.040 59.360 1490.38 642.56 12.96 11.23

 2005 187.030 89.340 1731.13 731.74 13.04 11.40

 2006 248.600 145.810 2069.34 820.30 13.11 11.57

 2007 386.500 240.110 2651.26 1055.14 13.18 11.74

 2008 518.400 315.850 3413.59 1027.91 13.25 11.91

 2009 433.830 296.560 3748.93 1126.95 13.31 12.08

 2010 617.610 409.150 4432.96 1375.39 13.38 12.25

 2011 739.080 505.370 5444.79 1488.52 13.44 12.41

 2012 687.900 539.400 6188.19 1489.24 13.51 12.37

 2013 654.700 484.400 6807.43 1498.87 13.57 12.52

 2014 706.050 542.260 7593.88 1595.70 13.64 12.95

 2015 716.200 582.400 7924.65 1581.59 13.71 13.11

 2016 755.400 678.900 8000.34 1599.00 13.72 13.22

 2017 809.998 599.450 8123.22 1601.22 13.98 13.56

 2018 825.445 601.220 8256.77 1598.88 14.31 13.89

 : China

 : Pakistan

2001 13.970 8.150 1041.64 490.04 12.72 1.48

 2002 17.995 12.420 1135.45 480.74 12.80 1.50

 2003 24.300 18.550 1273.64 543.59 12.88 1.53

 2004 30.610 24.660 1490.38 628.63 12.96 1.56

 2005 42.610 34.280 1731.13 690.85 13.04 1.59

 2006 52.460 42.390 2069.34 789.41 13.11 1.62

 2007 68.930 57.890 2651.26 870.63 13.18 1.64

 2008 70.570 60.510 3413.59 978.80 13.25 1.67

 2009 67.880 55.280 3748.93 949.12 13.31 1.70

 2010 86.680 69.370 4432.96 1018.87 13.38 1.74

 2011 105.570 84.390 5444.79 1194.33 13.44 1.77

 2012 124.130 92.750 6188.19 1290.36 13.51 1.79

 2013 142.150 110.190 6807.43 1299.12 13.57 1.82

 2014 160.060 132.480 7593.88 1334.15 13.64 1.85

 2015 189.300 164.500 7924.65 1428.99 13.71 1.89

 2016 219.000 189.000 8000.34 1521.80 13.72 1.99

 2017 225.340 210.990 8123.22 1788.90 13.98 1.97

 2018 279.900 235.790 8256.77 1899.90 14.31 1.89

 : China

 : Bangladesh

2001 9.720 9.550 1041.64 356.12 12.72 1.32

 2002 11.000 10.680 1135.45 354.30 12.80 1.34

 2003 13.700 13.360 1273.64 380.28 12.88 1.37

 2004 19.660 19.090 1490.38 407.99 12.96 1.39

 2005 24.800 24.020 1731.13 428.75 13.04 1.41

 2006 31.890 30.900 2069.34 434.84 13.11 1.42

 2007 34.590 33.450 2651.26 475.25 13.18 1.44

 2008 46.850 45.540 3413.59 546.85 13.25 1.45

 2009 45.820 44.410 3748.93 607.76 13.31 1.47

 2010 70.570 67.890 4432.96 674.93 13.38 1.49

 2011 82.600 78.100 5444.79 735.00 13.44 1.50

 2012 84.500 79.700 6188.19 747.34 13.51 1.55

 2013 103.070 97.050 6807.43 829.25 13.57 1.57

 2014 125.430 117.820 7593.88 1092.67 13.64 1.59

 2015 147.070 139.010 7924.65 1211.70 13.71 1.61

 2016 155.230 141.909 8000.34 1467.88 13.72 1.67

 2017 159.990 145.078 8123.22 1541.11 13.98 1.78

 2018 162.786 146.099 8256.77 1578.09 14.31 1.78

 : China

 : Afghanistan

2001 0.174 0.172 1041.64 92.21 12.72 0.267

 2002 0.199 0.199 1135.45 157.98 12.80 0.274

 2003 0.271 0.265 1273.64 168.68 12.88 0.282

 2004 0.579 0.569 1490.38 196.23 12.96 0.290

 2005 0.527 0.512 1731.13 227.88 13.04 0.299

 2006 1.006 1.004 2069.34 251.11 13.11 0.307

 2007 1.718 1.694 2651.26 306.98 13.18 0.316

 2008 1.543 1.516 3413.59 367.19 13.25 0.325

 2009 2.148 2.135 3748.93 425.07 13.31 0.334

 2010 1.789 1.752 4432.96 501.47 13.38 0.343

 2011 2.344 2.300 5444.79 575.97 13.44 0.353

 2012 4.692 4.640 6188.19 575.97 13.51 0.298

 2013 3.378 3.282 6807.43 678.35 13.57 0.305

 2014 4.109 3.393 7593.88 658.98 13.64 0.316

 2015 3.760 3.640 7924.65 590.27 13.71 0.325

 2016 3.980 3.780 8000.34 599.01 13.72 0.356

 2017 4.009 3.889 8123.22 600.89 13.98 0.339

 2018 4.145 3.990 8256.77 611.29 14.31 0.390

 : China

 : Bhutan

2001 0.0162 0.0160 1041.64 774.89 12.72 0.0588

 2002 0.0064 0.0062 1135.45 837.05 12.80 0.0606

 2003 0.0198 0.0197 1273.64 978.51 12.88 0.0624

 2004 0.0052 0.0035 1490.38 1074.58 12.96 0.0642

 2005 0.0047 0.0047 1731.13 1242.04 13.04 0.0659

 2006 0.0016 0.0016 2069.34 1330.52 13.11 0.0674

 2007 0.0539 0.0539 2651.26 1736.97 13.18 0.0688

 2008 0.0846 0.0846 3413.59 1792.91 13.25 0.0701

 2009 0.0417 0.0412 3748.93 1772.10 13.31 0.0713

 2010 0.0160 0.0159 4432.96 2088.49 13.38 0.0725

 2011 0.1746 0.1738 5444.79 2287.71 13.44 0.0738

 2012 0.1562 0.1560 6188.19 2398.91 13.51 0.0741

 2013 0.1741 0.1740 6807.43 2498.39 13.57 0.0753

 2014 0.1122 0.1112 7593.88 2380.91 13.64 0.0765

 2015 0.1531 0.1520 7924.65 2532.45 13.71 0.0774

 2016 0.1677 0.1677 8000.34 2667.89 13.72 0.0788

 2017 0.1998 0.1690 8123.22 2776.90 13.98 0.0791

 2018 0.2319 0.1709 8256.77 2890.00 14.31 0.0801

 : China

 : Sri Lanka

2001 3.967 3.866 1041.64 837.70 12.72 0.187

 2002 3.510 3.367 1135.45 903.90 12.80 0.189

 2003 5.240 5.040 1273.64 984.81 12.88 0.191

 2004 7.170 6.940 1490.38 1063.16 12.96 0.194

 2005 9.760 9.390 1731.13 1242.40 13.04 0.196

 2006 11.410 11.060 2069.34 1423.48 13.11 0.198

 2007 14.320 13.840 2651.26 1614.41 13.18 0.200

 2008 16.900 16.300 3413.59 2013.91 13.25 0.202

 2009 16.390 15.680 3748.93 2057.11 13.31 0.204

 2010 20.970 19.940 4432.96 2400.02 13.38 0.206

 2011 31.410 29.880 5444.79 2835.41 13.44 0.208

 2012 31.630 30.010 6188.19 2923.13 13.51 0.203

 2013 36.190 34.360 6807.43 3279.89 13.57 0.204

 2014 40.410 37.920 7593.88 3631.05 13.64 0.206

 2015 40.300 37.300 7924.65 3926.17 13.71 0.209

 2016 41.009 37.890 8000.34 4009.31 13.72 0.208

 2017 41.890 37.990 8123.22 4109.90 13.98 0.210

 2018 42.009 38.009 8256.77 4289.99 14.31 0.221

 : China

 : Maldives

2001 0.0220 0.0210 1041.64 2838.07 12.72 0.00277

 2002 0.0298 0.0296 1135.45 2885.49 12.80 0.00282

 2003 0.0335 0.0334 1273.64 3251.62 12.88 0.00286

 2004 0.0809 0.0791 1490.38 3638.57 12.96 0.00291

 2005 0.1696 0.1693 1731.13 3361.59 13.04 0.00295

 2006 0.1600 0.1540 2069.34 4353.01 13.11 0.00299

 2007 0.2500 0.2470 2651.26 5079.99 13.18 0.00303

 2008 0.3290 0.3150 3413.59 6149.01 13.25 0.00307

 2009 0.4080 0.4060 3748.93 6229.67 13.31 0.00311

 2010 0.6350 0.6340 4432.96 6570.43 13.38 0.00315

 2011 0.9730 0.9710 5444.79 6405.05 13.44 0.00320

 2012 0.7660 0.7640 6188.19 6566.65 13.51 0.00338

 2013 0.9780 0.9740 6807.43 6665.77 13.57 0.00345

 2014 1.0430 1.0390 7593.88 8483.81 13.64 0.00357

 2015 1.0920 1.0400 7924.65 7681.08 13.71 0.00409

 2016 1.1098 1.0448 8000.34 7767.89 13.72 0.00412

 2017 1.0989 1.0489 8123.22 7890.90 13.98 0.00467

 2018 1.0998 1.0509 8256.77 7998.12 14.31 0.00489

Table 3

Other initial data of the eight South Asian countries

 (kilometer)

 : China, : Nepal

1976.1 1 0 1 0

 : China, : India

2620.7 1 0 1 0

 : China, : Pakistan

2802.8 1 0 1 1

 : China, : Bangladesh 1874.6 0 0 1 0

 : China, : Afghanistan 3129.0 1 0 1 0

 : China, : Bhutan

1644.0 1 0 1 0

 : China, : Sri Lanka

4043.0 0 0 1 0

 : China, : Maldives

4711.0 0 0 1 0

In this paper we study the LKSARFT, so are the straight line distances between Lanzhou to the capital city of other related countries: Lanzhou-Kathmandu, Lanzhou-New Delhi, Lanzhou-Islamabad, Lanzhou-Dhaka, Lanzhou-Kabul, Lanzhou-Thimphu, Lanzhou-Colombo, Lanzhou-Male.

The results presented in Table 2 demonstrate a steady increase in the bilateral trade volume between China and the eight South Asian countries, as well as China's export volume to these nations from 2001 to 2018. Notably, the bilateral trade volume and export amount between China and India was the highest among the eight countries. The rankings of bilateral trade volume and export amount between China and the South Asian countries from one to eight are as follows: India, Pakistan, Bangladesh, Sri Lanka, Nepal, Afghanistan, Maldives, and Bhutan. Furthermore, the ranking of China's export amount to the eight South Asian countries from one to eight is also as follows: India, Pakistan, Bangladesh, Sri Lanka, Nepal, Afghanistan, Maldives, and Bhutan. The subsequent discussion will primarily focus on Nepal.

4. Results and discussion

4.1. The results and analysis

First of all, we should test whether the Eq.(6) and Eq.(7) are suitable to assess the performance of bilateral potential trade among the countries mentioned above. The testing results were presented in Table 4. We refused the assumptions because all LR Statistics values were larger than critical value, which means the two model are suitable to describe the non-efficiency trade, and the non-efficiency trade changes when time changes.

Table 4

The model testing results

LR statistics value Critical value (1%) Conclusion

Eq.(6) 

-173.2444 -299.4199 252.3511 11.34 Refuse

-299.4199 -391.4562 183.8726 6.63 Refuse

Eq.(7) 

-302.8503 -342.9575 80.2145 9.21 Refuse

-262.7431 -272.3425 19.1988 6.63 Refuse

We used Frontier 4.1 software to regression analyze the initial data mentioned in Table 2 and 3. The basic regression analysis and the model checking results were presented in Table 5. The trade efficiency analysis results between China and Nepal were presented in Table 6, including exports efficiency (EE) and bilateral trade efficiency (BTE), as well as the rankings of EE and BTE among the eight South Asian countries. The trade potential measurement results between China and Nepal were showed in Table 7.

Table 5

The basic regression analysis and model checking results based on SFGM

Independent variable Exports model Bilateral trade model

 Coefficient T statistic P value Coefficient T statistic P value

Constant term 8.1740 5.7900 0.0000 24.5357 24.8255 0.0000

0.2052 3.6128 0.0026 -0.0711 -0.6141 0.4635

2.0500 13.0621 0.0000 0.1449 0.6506 0.4356

7.2513 2.4249 0.0039 0.0329 0.0347 0.9268

-2.2414 -15.9151 0.0000 -2.0240 -7.3138 0.0000

-3.0095 -4.6072 0.0022 -0.0596 -0.6811 0.0001

0.5208 0.8756 0.4852 0.2101 0.2490 0.5486

-0.1630 -0.1650 0.8365 -1.7590 -10.3410 0.0000

5.0837 0.9458 0.3256 0.0127 0.0128 0.9325

1.2898 1.5898 0.1625 0.1515 0.1528 0.8735

2.2449 6.8936 0.0001 21.3774 21.8081 0.0000

0.9999 432.9989 0.0000 0.8433 17302.4610 0.0000

-2.5120 -4.8310 0.0019 0.0850 0.0850 0.9023

-0.0040 -0.1850 0.8269 0.0570 0.0670 0.9125

Log likelihood -173.2440 —— —— -262.7431 —— ——

LR Statistics 252.3511 —— —— 80.2145 —— ——

After analyzing the Table 5, we could obtain the following conclusions:

(i) For both the exports model and bilateral trade model, means non-efficiency trade’s influence on trade efficiency in two models are 84.33% and 99.99%, respectively. The exports model’s value is larger than the bilateral trade model, which means exports trade resistance from China to Nepal is larger than the bilateral trade resistance.

(ii) For both the exports model and bilateral trade model, shows that the non-efficiency factors existed in the trade processes, the SFGM is suitable to describe the trade between China and Nepal.

(iii) For the exports model and bilateral trade model, shows that the time-varying SFGMs are suitable to describe the trade between China and Nepal. For the bilateral trade model, shows the bilateral non-efficiency factor decreasing at a rate of 0.057 with time increasing, and the bilateral trade increasing at a rate of 0.057 with time increasing; For the exports model, means the exports non-efficiency factor increasing at a rate of 0.004 with time increasing, exports trade decreasing at a rate of 0.057 with time increasing.

(iv) For both the exports model and bilateral trade model, elasticity of per capita GDP in China is less than the value of the other seven trade countries, shows that the economic size of the trade countries has a great influence on China’s exports and bilateral trade. It is necessary to evaluate the seven trade countries’ economic development in the future if China wants more trades among the seven countries.

(v) The coefficients of the trade countries’ population are negative in the two models, indicates that if the trade countries have small number of population, they have smaller domestic markets, and they have smaller imports from China as well as bilateral trades. On the contrary, the coefficients of China’s population are negative in the two models; shows that the larger population constitutes a larger domestic market and larger imports demand form the trade countries. We thought that China’s labor force advantages have changed into the trade advantages.

(vi) The coefficients of the distance between two trade countries are negative in the two models. When the distance between two trading countries increased 1%, the exports trade value decreased 3.009%, and the bilateral trade value decreased 0.059%. Longer distance means larger transportation costs, which limits the development of trades.

(vii) The coefficients of and are negative in the exports model. Most of the countries sharing a common border with China are located in inner Asia-Europe Continent, the land transportation corridors development between two bordering countries are limited by the geographical environment, e.g., the mountains. Compared with the landlocked countries, China has both territorial land and territorial sea, the shipping provides a greater convenience for China’s exports trade.

(viii) The coefficients of are positive in the two models, English is the commonly used language among China and other counties while trading, it is helpful to improve the trade development if two trading countries share one common language.

(ix) The coefficients of are positive in the two models, which means the free trade agreement is critical to reduce bilateral trade barriers and trade non-efficiency factors.

Table 6

Exports efficiency (EE) and bilateral trade efficiency (BTE) results between China and Nepal based on SFGM

Year EE EE Rank BTE BTE Rank Year EE EE Rank BTE BTE Rank

2001 0.1324 4 0.0025 5 2010 0.0863 4 0.0124 5

2002 0.0744 4 0.0020 5 2011 0.1313 4 0.0179 5

2003 0.1053 4 0.0021 5 2012 0.1280 4 0.0322 5

2004 0.0837 4 0.0031 5 2013 0.1850 4 0.0338 5

2005 0.1035 4 0.0032 5 2014 0.1514 4 0.0378 5

2006 0.0883 4 0.0047 5 2015 0.1986 4 0.0434 5

2007 0.1379 4 0.0063 5 2016 0.2088 4 0.0452 5

2008 0.0688 4 0.0065 5 2017 0.2134 4 0.0466 5

2009 0.0959 4 0.0064 5 2018 0.2451 4 0.0478 5

The results in Table 6 show that larger efficiency value, larger EE and BTE, and smaller trade potential. Otherwise, smaller efficiency value, smaller EE and BTE, means larger trade potential. The BTE between China and Nepal increases when time changes, the EE from China to Nepal remains constant changing during the 18 years. The changing range of BTE is 0.002-0.05; the changing range of EE from China to Nepal is over 0.1, larger than the BTE. The BTE and EE ranking among the eight South Asian countries are ranking fifth and fourth during the 18 years (BTE ranking from 1 to 8: India, Pakistan, Bangladesh, Sri Lanka, Nepal, Afghanistan, Maldives, Bhutan; EE ranking from 1 to 8: India, Bangladesh, Pakistan, Nepal, Afghanistan, Bhutan, Sri Lanka, Maldives). The increasing BTE shows that the trade between China and Nepal has improved a lot; the trade potential has a lot of room to improve. Larger EE means the smaller exports trade potential, increasingly saturated exports markets from China to Nepal, some measures must be applied to improve China’s trade surplus.

The ratio of actual trade volume (EXP and EAI) and efficiency (EE and BTE, presented in Table 6) represents the Exports Trade Potential (ETP) and Bilateral Trade Potential (BTP), respectively. We could obtain the Extended Exports Trade Potential (EETP) and Extended Bilateral Trade Potential (EBTP) by substituting the original data (presented in Table 1 and Table 2) into the estimated random frontier gravity equations. The difference between ETP and EETP (BEP and EBTP) is, ETP (BEP) is influenced by the economic data of other seven countries, and EETP (EBTP) is the economic indicator which reflects the pure economic trade potential between China and Nepal. We subtracted 100% from the ratio of extended trade potential (EETP and EBTP) and trade potential (ETP and BTP), the results presented the Improved Exports Trade Potential (IETP) and Improved Bilateral Trade Potential (IBTP), respectively. All results were presented in Table 7.

4. The literature reviewed are inadequate, a lot of recent related studies are exempted from the review studies such as: Abdullahi et. al. (2022). Examining the determinants and efficiency of China’s agricultural exports using a stochastic frontier gravity model. Abdullahi, Aluko & Huo. (2021). Determinants, efficiency and potential of agri-food exports from Nigeria to the EU: Evidence from the stochastic frontier gravity model. Abdullahi et al. (2021). Determinants and Potential of Agri-Food Trade Using the Stochastic Frontier Gravity Model: Empirical Evidence From Nigeria. Jiang & Zhang (2021). Trade Sustainability and Efficiency under the Belt and Road Initiative: A Stochastic Frontier Analysis of China’s Trade Potential at Industry Level.

Author’s respond: We have added more descriptions about previous works in the literature review.

[1] used an extended gravity model to examine the determinants, efficiency and potential of agri-food exports from Nigeria to the EU for the 1995-2019 period the study showed that Nigeria’s agri-food exports with the EU has a relatively large potential that has not been exploited. The stochastic frontier gravity model is applied on a dataset including 35 countries during 2009–2017 by [24], the results indicated that the trade resistance of China’s export to countries along the Belt and Road has increased over time, while there is still huge trade potential at various industries. [2] provided empirical insights on the determinants and potential of agri-food exports from Nigeria to 70 major trading countries between 1995 and 2019 by applying a Stochastic Frontier Analysis on a gravity model. [3] aimed to examine the key determinants and efficiency of China’s agricultural exports with its 114 importing countries by applying the Stochastic Frontier Analysis on an augmented gravity model for the period of 2000–2019.

[1] Abdullahi, N. M., Aluko, O. A., Huo, X., 2021. Determinants, efficiency and potential of agri-food exports from nigeria to the eu: evidence from the stochastic frontier gravity model. Agricultural Economics (AGRICECON). 67(8), 337-349.

[2] Abdullahi, N. M., Huo, X., Zhang, Q., Azeez, A. B., 2021. Determinants and potential of agri-food trade using the stochastic frontier gravity model: empirical evidence from nigeria. SAGE Open. 11(4), 1-12.

[3] Abdullahi, N. M., Zhang, Q., Shahriar, S., Huo, X., 2022. Examining the determinants and efficiency of China’s agricultural exports using a stochastic frontier gravity model. PLoS ONE. 17(9), 0274187.

[24] Jiang, W., Zhang, H., Lin, Y., 2021. Trade sustainability and efficiency under the belt and road initiative: a stochastic frontier analysis of China's trade potential at industry level. Emerging Markets Finance and Trade. 58(1), 1-13.

Reviewer #2: 1. Title: The manuscript title is general and it is recommended to make it specific.

Author’s respond: We have revised the title.

Bilateral trade potential analysis of the Lanzhou-Kathmandu South Asian rail-road freight trains linking China and Nepal: A Stochastic Frontier Gravity Model approach.

2. Abstract: This section has been descriptively written. It is suggested to include the findings of the study in quantitate form here. In addition, policy implications should be included concisely at the end of this section.

Author’s respond: We have added the findings of the study in quantitate and policy implications in Abstract.

In this paper, the stochastic frontier gravity model is applied to analyze the trade potential between China and Nepal and the prospects of Lanzhou-Kathmandu South Asian rail-road freight trains (LKSARFT). Based on the statistical data, we test the Exports Efficiency (EE), Bilateral Trade Efficiency (BTE), Exports Trade Potential (ETP), Bilateral Trade Potential (BTP), Extended Exports Trade Potential (EETP), Extended Bilateral Trade Potential (EBTP), Improved Exports Trade Potential (IETP) and Improved Bilateral Trade Potential (IBTP) between China and Nepal, the following analysis results can be found: for the bilateral trade model, the bilateral non-efficiency factor decreasing at a rate of 0.057 with time increasing, bilateral trade increasing at a rate of 0.057 with time increasing. For the exports model, the exports non-efficiency factor increasing at a rate of 0.004 with time increasing, exports trade decreasing at a rate of 0.057 with time increasing. The BTE between China and Nepal increases when time changes, the EE from China to Nepal remains constant changing during the 18 years. The changing range of BTE is 0.002-0.05; the changing range of EE from China to Nepal is over 0.1, larger than the BTE. The BTE and EE ranking among the eight South Asian countries are ranking fifth and fourth during the 18 years. exports trade resistance from China to Nepal is larger than bilateral trade resistance; The import trade potential from Nepal to China is huge, the focus of bilateral trade between China and Nepal may be changed, there are more goods may be exported from Nepal to China, and China may become trade deficit when trading with Nepal. Then, the development bottlenecks of the LKSARFT are analyzed. Finally, we give policy directions to boost bilateral trade efficiency and tap the potential of bilateral trade between China and Nepal.

3. The manuscript is not properly organized in terms of (a). Introduction (b). Literature Review (c). Methodology of the study (d). Results and discussion (e). Conclusion and policy Implications.

Author’s respond: We have revised the organization of the text.

4. The novelty of the study is not properly outlined. It is recommended to include the novelty of the study at the end of the introduction.

Author’s respond: We have added the novelty of the study in the end of the introduction.

The advantages of studying the LKSARFT development prospects by applying SFGM are as follows: (i) SFGM does not suffer from loss of estimation efficiency. (ii) SFGM corrects for the economic distance bias term, which is creating non-normality, isolating it from the statistical error term. (iii) The suggested approach provides potential trade estimates that are closer to frictionless trade estimates, because the approach represents the upper limits of the data, which come from those economies that have liberalized their trade restrictions the most [32]. (iv) The SFGM bears strong theoretical and trade policy implications towards finding ways of minimizing unilateral impacts to volume of trade.

5. It is recommended to include a separate section on “Literature Review”. It is recommended to include in this section a review of previous studies directly or indirectly related to this study. Furthermore, it is recommended to find out the research gap that this study fills.

Author’s respond: We have included the separate section on Literature Review.

2.Literature Review

2.1. Bilateral trade potential: based on SFGM

Trade across regions and borders are considered important in improving welfare of people [7,19]. The B&R initiative provides good bilateral development opportunities for both China and Nepal, as well as the other seven South Asian countries. The LKSARFT acts as one of the important land belts, but how about the trade potential of the two countries? What factors promote or limit bilateral trade between the two countries? How to improve bilateral trade efficiency and tap the potential of bilateral trade between the two countries? All these questions’ quantitative study results are helpful to improve the development of the LKSARFT, as well as the development of the B&R. Next these questions are discussed based on stochastic frontier gravity model in detail.

2.2. The stochastic frontier gravity model

Some early literatures have estimated the difference between observed values and the estimated predicted values, by using an augmented gravity equation through Ordinary Least Squares (OLS) to assess the performance of bilateral potential trade among a pair of countries [6,36,15,5]. The OLS estimation procedure produces estimates that represent the centered values of the data set. However, potential trade refers to free trade with no restrictions to trade, some non-efficiency trade factors can’t be observed through this model, so for policy purposes, it is rational to define potential trade as a maximum possible trade that can occur between any two countries, which has liberalized trade restrictions the most, given the determinants of trade. This means that the estimation of the potential trade requires a procedure that represents the upper limits of the data and not the centered values of the data [27]. To solve this problem, [13] proposed the concept of stochastic production frontier analysis which deals with the upper bound of the data set to measure the maximum possible output, this approach is known as the Stochastic Frontier Gravity Model (SFGM). The SFGM is the Integration of Gravity Model and Stochastic Frontier Production Function Model which was formally introduced by [26] to address the inherent bias of the conventional gravity model of trade and to estimate potential trade flows. [53] applied frontier production function analysis to small farms in Nicaragua during 1998-2005, the results showed an acceptable average of technical efficiency which the makers of public policy in Nicaragua a must consider for the future. [40] focused on examining Philippines’s exports efficiency and potential based on trading partner’s characteristics using SFGM, unlike the usual measure of gravity model using OLS that measure potential from the mean. [12] provided a survey about environmental efficiency measure issue. the paper deals with different ways for including environmental variables, which offered several perspectives for measuring efficiency with frontier methods. [45] estimated the potential and efficiency of trade flows between China and countries along the Maritime Silk Road from 2005 to 2013 by using the SFGM. Furthermore, [17] defined the extent of the Belt and Road in terms of geographical boundaries, justified the application of the SFGM to the foreign direct investment (FDI) analysis, and constructed a frontier regression model to assess the China’s outward FDI efficiency. [1] used an extended gravity model to examine the determinants, efficiency and potential of agri-food exports from Nigeria to the EU for the 1995-2019 period the study showed that Nigeria’s agri-food exports with the EU has a relatively large potential that has not been exploited. The stochastic frontier gravity model is applied on a dataset including 35 countries during 2009–2017 by [24], the results indicated that the trade resistance of China’s export to countries along the Belt and Road has increased over time, while there is still huge trade potential at various industries. [2] provided empirical insights on the determinants and potential of agri-food exports from Nigeria to 70 major trading countries between 1995 and 2019 by applying a Stochastic Frontier Analysis on a gravity model. [3] aimed to examine the key determinants and efficiency of China’s agricultural exports with its 114 importing countries by applying the Stochastic Frontier Analysis on an augmented gravity model for the period of 2000–2019.

6. It is recommended that a separate section on “methodology” should include (a). a table regarding “description of the variables” used in this study. (b). It is recommended to discuss the significance of each variable included in this study/model (c). Methods for measuring trading potential between trading partners should be included. (d). It is recommended to include the econometric model that examines the impact of factors including BRI that affect trading potential.

Author’s respond: We have added the description of the variables in section 3.

Table 1 The variables

Variable Explanation Variable Explanation

The bilateral trade volume between country and country 

Exports volume from country to country 

The per capita gross domestic product in country 

The total population in country 

The per capita gross domestic product in country 

The total population in country 

The distance between the two countries 

Situation of border between the two countries

The type of land between the two countries 

Type of language between the two countries

Also, we have discussed methods for measuring trading potential, please see the discussion in section 4.

The ratio of actual trade volume (EXP and EAI) and efficiency (EE and BTE, presented in Table 6) represents the Exports Trade Potential (ETP) and Bilateral Trade Potential (BTP), respectively. We could obtain the Extended Exports Trade Potential (EETP) and Extended Bilateral Trade Potential (EBTP) by substituting the original data (presented in Table 2 and Table 3) into the estimated random frontier gravity equations. The difference between ETP and EETP (BEP and EBTP) is, ETP (BEP) is influenced by the economic data of other seven countries, and EETP (EBTP) is the economic indicator which reflects the pure economic trade potential between China and Nepal. We subtracted 100% from the ratio of extended trade potential (EETP and EBTP) and trade potential (ETP and BTP), the results presented the Improved Exports Trade Potential (IETP) and Improved Bilateral Trade Potential (IBTP), respectively. All results were presented in Table 7.

Table 7

Trade potential measurement results between China and Nepal (Billion USD)

Year Exports trade potential from China to Nepal Bilateral trade potential

 EXP ETP EETP IETP EAI BTP EBTP IBTP

2001 1.486 11.226 11.362 1.20% 1.532 611.131 873.801 42.98%

2002 1.051 14.136 18.847 33.33% 1.103 556.352 759.760 36.56%

2003 1.220 11.590 11.146 -3.83% 1.273 607.900 796.939 31.10%

2004 1.632 19.489 24.563 26.03% 1.715 561.220 703.717 25.39%

2005 1.879 18.162 16.521 -9.03% 1.964 621.060 743.909 19.78%

2006 2.598 29.419 32.576 10.73% 2.680 567.938 615.128 8.31%

2007 3.860 27.982 24.368 -12.92% 4.000 631.958 698.734 10.57%

2008 3.750 54.532 61.257 12.33% 3.810 586.660 622.423 6.10%

2009 4.090 42.652 35.940 -15.74% 4.140 643.052 662.756 3.06%

2010 7.322 84.851 91.225 7.51% 7.437 600.222 594.806 -0.90%

2011 11.810 89.917 72.541 -19.32% 11.950 667.626 638.855 -4.31%

2012 19.680 153.796 208.593 35.63% 19.980 620.408 733.035 18.15%

2013 22.100 119.461 123.302 3.22% 22.540 666.582 769.951 15.51%

2014 22.830 150.808 200.647 33.05% 23.300 617.088 693.291 12.35%

2015 27.890 140.424 142.450 1.44% 28.660 659.878 724.848 9.85%

2016 28.900 141.234 144.900 2.60% 30.245 689.998 700.909 1.58%

2017 31.008 150.900 157.770 4.55% 32.335 699.908 711.789 1.70%

2018 33.450 151.990 153.890 1.25% 35.667 710.990 767.897 8.00%

The results in Table 7 show that the BTP is larger than ETP between China and Nepal, also we find that the average IBTP from 2001 is over 15%, but the average IETP from 2001 is just 6%, in 2003, 2005, 2007, 2009 and 2011 the IETP values are negative, which means the import trade potential from Nepal to China is huge, the focus of bilateral trade between China and Nepal may be changed, there are more goods may be exported from Nepal to China, and China may become trade deficit when trading with Nepal. Absolutely, the LKSARFT provides a convenient, safe, time-saving and cost-saving land transport corridor in the trades between China and Nepal.

Since all the data used in this study are collected historical data, which are fixed, and the dimensions of the data are very different, it is troublesome to analyze the importance of variables. Moreover, this study mainly studies the influence of variables on trade potential, rather than analyze variables separately. However, we will discuss the significance of each variable and examine impact of factors that affect trading potential in the future. In response to this, we have added more descriptions of these jobs in the future works part.

Some basic problems, such as operational and management bottlenecks still remained unresolved. How to solve the questions mentioned above, are the further study works; furthermore, distance is the key variable in this paper, and it reflects the transportation cost between two countries and inversely proportional to trade, and the traffic routes and transport modes between China and South Asian Countries are different, so next we will try to improve our work using the distance (between ports, main trading city or capital) multiplied by freight rate (ocean freight, road freight or rail freight) to analyze the bilateral trade between two countries; LKSARFT may reduce the transportation cost between China and Nepal, and even promote trade between China and other South Asian Countries, the new transport lines of LKSARFT are under construction, so next we can also introduce new distance reduced by LKSARFT in the SFGM to predict the trade potential under LKSARFT. Moreover, in order to refine the study, we will discuss the significance of each variable and examine impact of factors that affect trading potential in the future.

7. Referred to Table 1, it is recommended to include descriptive statistics for the data series rather than including the full data.

Author’s respond: We have added descriptive statistics for the data series in section 3. Please see the red part.

In this paper we study the LKSARFT, so are the straight line distances between Lanzhou to the capital city of other related countries: Lanzhou-Kathmandu, Lanzhou-New Delhi, Lanzhou-Islamabad, Lanzhou-Dhaka, Lanzhou-Kabul, Lanzhou-Thimphu, Lanzhou-Colombo, Lanzhou-Male.

The results presented in Table 2 demonstrate a steady increase in the bilateral trade volume between China and the eight South Asian countries, as well as China's export volume to these nations from 2001 to 2018. Notably, the bilateral trade volume and export amount between China and India was the highest among the eight countries. The rankings of bilateral trade volume and export amount between China and the South Asian countries from one to eight are as follows: India, Pakistan, Bangladesh, Sri Lanka, Nepal, Afghanistan, Maldives, and Bhutan. Furthermore, the ranking of China's export amount to the eight South Asian countries from one to eight is also as follows: India, Pakistan, Bangladesh, Sri Lanka, Nepal, Afghanistan, Maldives, and Bhutan. The subsequent discussion will primarily focus on Nepal.

8. It is recommended to justify why the author used the SFGM model instead of other econometric models.

Author’s respond: We have added the advantages of studying the LKSARFT development prospects by applying SFGM in the end of the introduction.

The advantages of studying the LKSARFT development prospects by applying SFGM are as follows: (i) SFGM does not suffer from loss of estimation efficiency. (ii) SFGM corrects for the economic distance bias term, which is creating non-normality, isolating it from the statistical error term. (iii) The suggested approach provides potential trade estimates that are closer to frictionless trade estimates, because the approach represents the upper limits of the data, which come from those economies that have liberalized their trade restrictions the most [32]. (iv) The SFGM bears strong theoretical and trade policy implications towards finding ways of minimizing unilateral impacts to volume of trade.

9. It is recommended to correlate your study with previous studies conducted. In addition, it is recommended to prescribe the policy implications based on the study results in a separate paragraph after the study conclusions.

Author’s respond: We have prescribed the policy implications based on the study results in a separate paragraph after the study conclusions. Please see the Penultimate paragraph.

In line with these findings, we give policy directions to boost bilateral trade efficiency and tap the potential of bilateral trade between the two countries: (i) Export goods would be stimulated by formulating and implementing macro-economic policies aimed at increasing the economic size of China. (ii) The establishment of a robust and comprehensive integrated process for foreign exports would facilitate the process of foreign exports, consequently reducing China's export trade impediments with Nepal. (iii) In order to mitigate transportation costs, China should establish and implement tax incentive policies for cross-border trade in encouraged sectors. (iv) Both countries should promote the teaching of each other's languages in their respective education systems to enhance communication and cooperation, and facilitate cross-cultural understanding, thereby promoting trade development between the two nations. (iv) To rectify China's trade surplus, it is imperative for the nation to calibrate its fiscal policies such that there is a surge in demand, thereby inducing an increase in import volume.

10. It is recommended to enrich the manuscript with a literature review.

Author’s respond: We have included the separate section on Literature Review.

2. Literature Review

2.1 Bilateral trade potential: based on SFGM

Trade across regions and borders are considered important in improving welfare of people [7,19]. The B&R initiative provides good bilateral development opportunities for both China and Nepal, as well as the other seven South Asian countries. The LKSARFT acts as one of the important land belts, but how about the trade potential of the two countries? What factors promote or limit bilateral trade between the two countries? How to improve bilateral trade efficiency and tap the potential of bilateral trade between the two countries? All these questions’ quantitative study results are helpful to improve the development of the LKSARFT, as well as the development of the B&R. Next these questions are discussed based on stochastic frontier gravity model in detail.

2.2 The stochastic frontier gravity model

Some early literatures have estimated the difference between observed values and the estimated predicted values, by using an augmented gravity equation through Ordinary Least Squares (OLS) to assess the performance of bilateral potential trade among a pair of countries [6,36,15,5]. The OLS estimation procedure produces estimates that represent the centered values of the data set. However, potential trade refers to free trade with no restrictions to trade, some non-efficiency trade factors can’t be observed through this model, so for policy purposes, it is rational to define potential trade as a maximum possible trade that can occur between any two countries, which has liberalized trade restrictions the most, given the determinants of trade. This means that the estimation of the potential trade requires a procedure that represents the upper limits of the data and not the centered values of the data [27]. To solve this problem, [13] proposed the concept of stochastic production frontier analysis which deals with the upper bound of the data set to measure the maximum possible output, this approach is known as the Stochastic Frontier Gravity Model (SFGM). The SFGM is the Integration of Gravity Model and Stochastic Frontier Production Function Model which was formally introduced by [26] to address the inherent bias of the conventional gravity model of trade and to estimate potential trade flows. [53] applied frontier production function analysis to small farms in Nicaragua during 1998-2005, the results showed an acceptable average of technical efficiency which the makers of public policy in Nicaragua a must consider for the future. [40] focused on examining Philippines’s exports efficiency and potential based on trading partner’s characteristics using SFGM, unlike the usual measure of gravity model using OLS that measure potential from the mean. [12] provided a survey about environmental efficiency measure issue. the paper deals with different ways for including environmental variables, which offered several perspectives for measuring efficiency with frontier methods. [45] estimated the potential and efficiency of trade flows between China and countries along the Maritime Silk Road from 2005 to 2013 by using the SFGM. Furthermore, [17] defined the extent of the Belt and Road in terms of geographical boundaries, justified the application of the SFGM to the foreign direct investment (FDI) analysis, and constructed a frontier regression model to assess the China’s outward FDI efficiency. [1] used an extended gravity model to examine the determinants, efficiency and potential of agri-food exports from Nigeria to the EU for the 1995-2019 period the study showed that Nigeria’s agri-food exports with the EU has a relatively large potential that has not been exploited. The stochastic frontier gravity model is applied on a dataset including 35 countries during 2009–2017 by [24], the results indicated that the trade resistance of China’s export to countries along the Belt and Road has increased over time, while there is still huge trade potential at various industries. [2] provided empirical insights on the determinants and potential of agri-food exports from Nigeria to 70 major trading countries between 1995 and 2019 by applying a Stochastic Frontier Analysis on a gravity model. [3] aimed to examine the key determinants and efficiency of China’s agricultural exports with its 114 importing countries by applying the Stochastic Frontier Analysis on an augmented gravity model for the period of 2000–2019.

We sincerely thank all reviewers and editors for your valuable advices and comments on this paper, because the advices and comments are great helpful to improve this article. If there are any other problems and questions need to be handled in this paper, please tell us, thank you so much!

---

## [Decision Letter · Decision Letter 1]

20 Apr 2023

Bilateral trade potential analysis of the Lanzhou-Kathmandu South Asian rail-road freight trains linking China and Nepal: A Stochastic Frontier Gravity Model approach

PONE-D-23-03918R1

Dear Dr. Fei Tian,

We’re pleased to inform you that your manuscript has been judged scientifically suitable for publication and will be formally accepted for publication once it meets all outstanding technical requirements.

Kind regards,

C. A. Zúniga-González, Ph.D

Academic Editor

PLOS ONE

Additional Editor Comments (optional):

Dear authors, I am checked that all reviewers observatios were made, my congratulations. My decsion is accept.

Reviewers' comments:

Reviewer's Responses to Questions

**Comments to the Author**

1. If the authors have adequately addressed your comments raised in a previous round of review and you feel that this manuscript is now acceptable for publication, you may indicate that here to bypass the “Comments to the Author” section, enter your conflict of interest statement in the “Confidential to Editor” section, and submit your "Accept" recommendation.

Reviewer #1: All comments have been addressed

Reviewer #2: All comments have been addressed

2. Is the manuscript technically sound, and do the data support the conclusions?

Reviewer #1: Yes

Reviewer #2: Yes

3. Has the statistical analysis been performed appropriately and rigorously? 

Reviewer #1: Yes

Reviewer #2: Yes

4. Have the authors made all data underlying the findings in their manuscript fully available?

Reviewer #1: Yes

Reviewer #2: Yes

5. Is the manuscript presented in an intelligible fashion and written in standard English?

Reviewer #1: Yes

Reviewer #2: Yes

6. Review Comments to the Author

Reviewer #1: The authors have adequately addressed all the previous comments. The manuscript is now suitable for publication in this journal.

Reviewer #2: Authors have addressed all my comments. I have no further comments. I recommend the manuscript for publication.

7. PLOS authors have the option to publish the peer review history of their article (what does this mean?). If published, this will include your full peer review and any attached files.

Reviewer #1: **Yes: **Nazir Muhammad Abdullahi

Reviewer #2: **Yes: **Prof. Dr. Dilawar Khan, Department of Economics, Kohat University of Science & Technology

---

## [Editor Report · Acceptance letter]

28 Apr 2023

PONE-D-23-03918R1 

Bilateral trade potential analysis of the Lanzhou-Kathmandu South Asian rail-road freight trains linking China and Nepal: A Stochastic Frontier Gravity Model approach 

Dear Dr. Tian:

I'm pleased to inform you that your manuscript has been deemed suitable for publication in PLOS ONE. Congratulations! Your manuscript is now with our production department. 

Kind regards, 

on behalf of

Dr. Prof. C. A. Zúniga-González 

Academic Editor

PLOS ONE